# Benchmarking Heterogeneous Treatment Effect Models through the Lens of Interpretability

**Jonathan Crabbé**[*]
University of Cambridge
jc2133@cam.ac.uk

**Alicia Curth**[*]
University of Cambridge
amc253@cam.ac.uk

**Ioana Bica**[*]
University of Oxford
The Alan Turing Institute
ioana.bica@eng.ox.ac.uk

**Mihaela van der Schaar**
University of Cambridge
The Alan Turing Institute
UCLA
mv472@cam.ac.uk

## Abstract

Estimating personalized effects of treatments is a complex, yet pervasive problem. To tackle it, recent developments in the machine learning (ML) literature on heterogeneous treatment effect estimation gave rise to many sophisticated, but opaque, tools: due to their flexibility, modularity and ability to learn constrained representations, neural networks in particular have become central to this literature. Unfortunately, the assets of such black boxes come at a cost: models typically involve countless nontrivial operations, making it difficult to *understand* what they have learned. Yet, understanding these models can be crucial – in a medical context, for example, discovered knowledge on treatment effect heterogeneity could inform treatment prescription in clinical practice. In this work, we therefore use post-hoc *feature importance* methods to identify features that influence the model's predictions. This allows us to evaluate treatment effect estimators along a new and important dimension that has been overlooked in previous work: We construct a benchmarking environment to empirically investigate the ability of personalized treatment effect models to identify *predictive covariates* – covariates that determine differential responses to treatment. Our benchmarking environment then enables us to provide new insight into the strengths and weaknesses of different types of treatment effects models as we modulate different challenges specific to treatment effect estimation – e.g. the ratio of prognostic to predictive information, the possible nonlinearity of potential outcomes and the presence and type of confounding.

## 1  Introduction

The need to estimate the effects of actions – such as treatments, policies and other interventions – is ubiquitous in many domains, ranging from economics to medicine. Many applications where treatment effects are of interest additionally operate under *high stakes*, for example treatment decisions in a hospital setting – making it particularly important that estimates leading to individual decisions are reliable. As interest in designing *personalized* treatments is growing across fields, a substantial literature on learning treatment effect heterogeneity has emerged in machine learning (ML) in recent years. In this context, a plethora of sophisticated methods for estimating *conditional average treatment*

---

[*]Equal contribution.

36th Conference on Neural Information Processing Systems (NeurIPS 2022) Track on Datasets and Benchmarks.

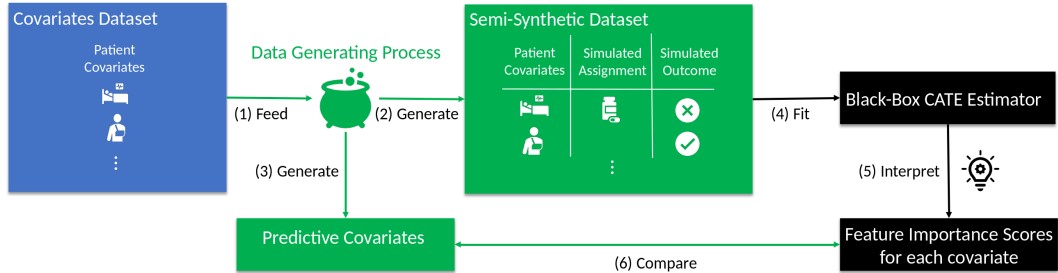

Figure 1: Illustration of the ITErpretability benchmarking environment. (1) Covariates are extracted from any dataset. (2) These covariates are labeled with a transparent data generating process for the assignments and the outcomes. This results in a semi-synthetic dataset $\mathcal{D}$. (3) Since the data generating process is transparent, we know the indices $\mathcal{I}_{\mathrm{pred}}$ of predictive covariates. (4) We fit a black-box CATE estimator $\hat{\tau}$ on $\mathcal{D}$. (5) We then use a feature importance method to assign feature importance scores $a_i(\hat{\tau}, x)$ to each covariate $x_i$. (6) We evaluate each CATE estimator based on a new metric measuring the accordance between the most important features and the predictive covariates.

*effects* (CATE) and/or an individual's *potential outcomes* (POs) under different treatments have been proposed in the recent ML literature (see e.g. [1]), all aiming to improve the *precision* in estimating effects. In particular, recent work has produced both *model-agnostic* estimation strategies, which can incorporate *any* ML method into the estimation of effects [2–4], *and* model-specific strategies, which *adapt specific* ML methods to the treatment effect estimation problem – for example random forests [5, 6], gaussian processes [7, 8] and, most predominantly, neural networks [4, 9–13].

Despite the growing interest in such methods, [14] noted that their evaluation has been quite one-dimensional: Most, if not all, of this work has focussed on assessing performance of proposed algorithms in terms of the *Precision of Estimating Heterogeneous Effects* (PEHE) criterion of [15], which measures the root mean squared error (RMSE) of the estimated CATE function on a test-set. However, in many applications, black-box predictions of expected treatment effects do not suffice. In the context of drug development, for example, it is at least equally important to assess whether an algorithm *discovers the correct drivers of the underlying effect heterogeneity* or leads to the right *interpretation* thereof [16]: discovering effect-modifiers can be important both in exploratory clinical trial phases, e.g. to identify potential biomarkers which may help to explain an enhanced treatment effect for future drug development, and in confirmatory trial phases, e.g. to rule out that a drug is ineffective for some biomarker signatures or to otherwise adapt prescription criteria [16, 17]. Such interpretation could also be important in clinical practice e.g. when *explaining* treatment recommendations derived from an estimated CATE function. In this paper, we therefore leverage recent advances in *explainable artificial intelligence* [18–20] to interpret the discoveries of ML-based CATE estimators, and propose a benchmarking environment which we use to provide insight into the performance of different learning strategies in discovering drivers of heterogeneity.

**Related work.** How to interpret CATE estimators has received little attention in the ML literature thus far. Some work *implicitly* enables interpretation by relying on methods that are inherently more interpretable, e.g. linear regressions [21, 22] or tree-based models [23, 24]. Another recent stream of work *explicitly* focuses on interpretability, similarly proposing the use of methods that are inherently interpretable: [25] rely on decision lists, [26] construct interpretable hyper-boxes for matching, [27] use causal rule ensembles, [28] use mixture models with sparse components, [29] use the fused lasso for estimation of subgroup piece-wise constant treatment effects and [30] rely on an additive neural network architecture and [31] propose using transformer backbones that can be interpreted using their attention weights. With a goal similar to our work, [16] investigate how well different *forest-based* CATE estimation strategies discover effect modifiers, but they rely on variable importance scores inherent to random forests to do so. To the best of our knowledge, the only work that considers *post-hoc* interpretability of already fitted, arbitrary black-box CATE models uses *model distillation* approaches to do so: [32] use the fitted black-box CATE estimator as a "teacher" and a multi-task decision tree as an interpretable "student" and similarly, [33] propose to create interpretable CATE estimators by fitting an arbitrary interpretable model on top of the potential outcome predictions of a NN-based first stage estimator. Such approaches do not provide an interpretation of the black-box output directly and therefore need to rely on the interpretable student model being good enough to cover the complexities of the original estimator. In this paper, we take a different approach and consider the use of *post-hoc feature importance methods* to interpret black-box CATE estimators *directly*.

**Contributions & Outlook.** Our contributions are threefold: (i) we study how to *interpret* black-box CATE estimators and use this to *evaluate* them along a *new and important dimension* that has been overlooked in previous work – namely, their ability to correctly discover drivers of effect heterogeneity –, (ii) we propose a *benchmark environment* to do so, and (iii) we provide *new insights* into the performance of existing methods on this new task. We proceed as follows: We begin by recalling fundamentals of the CATE estimation setting and discuss its unique characteristics in Section 2. In Section 3, we review feature importance methods and discuss how they can be applied to interpret what CATE estimators have learned about treatment effect heterogeneity, with the ultimate goal to *compare different CATE estimators* on their ability to identify predictive covariates/features (i.e. covariates that determine differential responses to treatment and, hence, are the ones that truly matter when estimating treatment effects). In Section 4, we then introduce a new benchmark environment for evaluating black-box CATE estimators, supplemented with feature importance methods to interpret their output, on precisely this task. This *ITErpretability* benchmark, as illustrated in Figure 1, can be used with any covariates dataset and is semi-synthetic: it relies on real covariates with synthetic treatment assignment and outcomes, supplying us with otherwise unavailable *ground-truth access* to predictive covariates. Finally, in Section 5, we use this benchmark to investigate the performance of existing, neural network-based, CATE estimation strategies on this new task, and provide interesting insights into the impact of typical difficulties in CATE problems: the strength of predictive relative to prognostic information, the possible nonlinearity of potential outcomes and the presence of confounding.

## 2   Setting: The CATE Estimation Problem

We consider a standard treatment effect estimation setting as formalized within the Neyman-Rubin potential outcomes framework [34]. We assume access to an observational i.i.d. dataset $\mathcal{D} = \{(Y^n, X^n, W^n)\}_{n=1}^N$ Here, $Y^n \in \mathcal{Y} \subset \mathbb{R}$ is a binary or continuous outcome of interest, $X^n \in \mathcal{X} \subset \mathbb{R}^d$ is a vector of pre-treatment covariates, and $W^n \in \{0,1\}$ is a binary treatment assigned according to a (usually unknown) propensity score $\pi(x) = P(W = 1 \mid X = x)$. Each individual has multiple potential outcomes (POs) $Y^n(w)$ associated with different treatment values, yet only the outcome associated with the received treatment is observed, i.e. $Y^n = Y^n(W^n)$. To investigate treatment effect heterogeneity, we focus on the conditional average treatment effect (CATE), computed as the expected difference between the two POs for an individual with covariates $X = x$:

$$\tau(x) = \mathbb{E}_P[Y(1) - Y(0) \mid X = x] = \mu_1(x) - \mu_0(x) \tag{1}$$

with $\mu_w(x) = \mathbb{E}_P[Y(w) \mid X = x]$ the expected potential outcome.

**What Makes CATE Estimation Special Relative to Standard Supervised Learning?** As we discuss in more detail in Appendix A, three characteristics are generally considered central to the CATE estimation problem (see e.g. [13, 14]): ① The need to rely on strong untestable assumptions to ensure *identifiability* of treatment effects from observational data (here: ignorability assumptions [35], as discussed further in Appendix A). ② The presence of *covariate shift* due to confounding (correlation between covariates and treatment assignment). ③ The *absence of the target label* of interest $(Y(1) - Y(0))$ as only one of the POs can be observed in practice, or conversely, that CATE can arise either as a single regression or as a *difference* between two functions

In our context, ① & ② are mainly important for the *construction of benchmarks*: as discussed in Section 4, characteristic ① (and to some extent ③) leads to the necessity to rely on (semi-)*synthetic* data as ground truth knowledge of the data-generating process is generally unavailable from real data (see also the discussion in [14]). Characteristic ② leads to a natural *experimental knob* to include in a benchmark to be modulated when evaluating models. As we discuss in detail in Section 3, it is ③ that leads to most interesting considerations when studying *how to interpret* CATE estimators. It also lead to the emergence of different CATE estimation strategies: generally, as discussed in [4], $\tau(x)$ can either be estimated *indirectly* using the difference between the PO regression estimates (i.e. as $\hat{\mu}_1 - \hat{\mu}_0$) as in most methods proposed in the recent ML literature [7, 8, 10, 12, 13] or *directly* by fitting a regression model using pseudo-outcome $Y_{\hat{\eta}}$ as a surrogate for the unobserved PO difference (relying on initial estimates of some of the nuisance functions $\eta = (\mu_0, \mu_1, \pi)$) as in e.g. [2–4]. We discuss different instantiations of these strategies considered in our experiments in Section 5.

# 3 Feature Importance in the Context of Treatment Effect Estimation

In this section, we discuss how to use feature importance to investigate what a CATE estimator (or: model) has learned about treatment effect heterogeneity. We start by describing a general feature importance formalism: We assume that we have access to a *black-box* model $\hat{\tau} : \mathbb{R}^d \to \mathbb{R}$ to estimate the CATE $\tau$. Feature importance methods permit to understand the prediction of this model by highlighting features (covariates) the model is sensitive to. Concretely, this is done by assigning an importance score $a_i(\hat{\tau}, x) \in \mathbb{R}$ to each feature $x_i$ contained in the vector $x$ with [2] $i \in [d]$. This score reflects the importance of $x_i$ to predict the CATE $\hat{\tau}(x)$. The score is such that the importance of a feature $x_i$ increases with $|a_i(\hat{\tau}, x)|$. When it comes to the sign, features with $a_i(\hat{\tau}, x) > 0$ tend to increase the CATE $\hat{\tau}(x)$ and features with $a_i(\hat{\tau}, x) < 0$ tend to decrease it.

There exist many methods to assign importance scores $a_i(\hat{\tau}, x)$ to the features. Different feature importance methods tend to attribute different relative importance between features [36, 37]. This is because they measure different characteristics of the black-box model: *gradient-based* methods compute scores based on the model's gradient with respect to the features [38–41]; *perturbation-based* methods compute scores based on the model's sensitivity to features perturbations [42–44] and some other methods rely on the neuron's activation to compute the importance scores [45, 46]. In Section 5, we use Integrated Gradients [39] as our main feature importance method. This is because it offers the best performances empirically and is typically more computationally efficient than the previous methods. A comparison between different feature importance methods is provided in Appendix B. We will now discuss the specificity of CATE models in an interpretability perspective.

**What Makes CATE Interpretability Special?** Above, we fixed the black-box to interpret to be the CATE estimate $\hat{\tau}$. However, CATE can of course also be written (and, as in most methods proposed in the recent ML literature, be estimated) in terms of the expected POs, i.e. $\tau = \mu_1 - \mu_0$, which have their own feature importance $a_i(\mu_0, x)$ and $a_i(\mu_1, x)$. In general, feature importance for CATE will differ from those of the POs, leading to different insights. This multiplicity of interpretation certainly distinguishes CATE models from interpreting standard supervised learners and deserves a discussion.

In order to attach meaning to those possible interpretations, it is useful to consider an important distinction between so-called *predictive* and *prognostic* covariates made in the medical literature [47, 48]. Prognostic covariates determine outcome regardless of treatment assignment – common risk factors such as age or gender may fall in this category. Such variables are taken into account by the two potential outcomes $\mu_w$ in a similar way. In this way, prognostic covariates correspond to features $x_i$ that are important for both POs $a_i(\mu_0, x) \neq 0$ and $a_i(\mu_1, x) \neq 0$. Predictive covariates, on the other end, are predictive of effect heterogeneity, i.e they determine differential responses to treatment – hormone receptor status in cancer patients is one such example [49]. In this way, predictive covariates correspond to features $x_i$ that are important for the CATE $a_i(\tau, x) \neq 0$. If we assume that the potential outcomes $\mu_0, \mu_1$ are differentiable with respect to the covariates, one can extend the previous definitions globally. In this case, a feature $x_i$ is *prognostic* if both potential outcomes depend on that feature: $\partial\mu_0/\partial x_i \neq 0$ and $\partial\mu_1/\partial x_i \neq 0$. Similarly, a feature $x_i$ is *predictive* if the CATE depends on that feature: $\partial\tau/\partial x_i \neq 0$. In the medical context, any measured patient covariate could be prognostic, predictive or both (or, of course, irrelevant).

Whereas [33] used a model-distillation approach to interpret only the PO models, we argue that it is most interesting in the CATE estimation context to interpret the learned CATE models *directly* by focusing on the discovery of predictive covariates. Identifying such predictive covariates can, for example, provide precious information to support exploratory analyses in clinical trials, which in turn allow pharmaceutical companies to refine the target population for a treatment, improving the likelihood of a successful later stage trial [16]. From that perspective, CATE models $\hat{\tau}$ that are better at identifying predictive covariates are clearly preferable. Due to the absence of treatment effect labels (Section 2), it is far from obvious whether all estimation strategies result in successful identification of predictive covariates. Furthermore, since indirect learners target the CATE indirectly through the POs, there is no guarantee that the resulting models can distinguish between prognostic and predictive covariates. With these simple observations, it is obvious that CATE interpretability comes with a unique set of challenges. We will now introduce a benchmark to study CATE models through this angle.

---

[2]In the following, $[n]$ denotes the set of natural numbers between 1 and $n \in \mathbb{N}^*$.

# 4 The ITErpretability Benchmark

Next, we describe our proposed *ITErpretability* benchmark that uses ideas from *interpretability* to measure the ability of *treatment effect* estimators to identify predictive covariates. We propose a framework that relies on a *semi-synthetic* data generating process (DGP), which is standard in the CATE estimation literature [14]: because identifiability assumptions are generally untestable, *simulating* outcomes and treatment assignments ensures that they hold; additionally it ensures that the underlying CATE function is *known*. In our context, we cannot rely on existing and established semi-synthetic benchmarks such as IHDP [15, 10] or ACIC2016 [50], most importantly because they did not *record* which covariates are predictive or prognostic. Further, the *experimental knobs* considered therein are not of primary interest in our setting[3]; instead we thus design our own DGPs that allow to us to obtain interesting new insights in our experiments. Below, we discuss our DGP and proposed metrics.

**DGP.** We would like to rely on a DGP that covers a range of realistic scenarios *and* for which we can clearly identify prognostic and predictive covariates. Since this last information is generally not available in *real* observational data, we use a semi-synthetic approach in which we reuse covariates $X^n$ from a real dataset and synthetically generate the treatment assignments $W^n$ and outcomes $Y^n$ for all $n \in [N]$. In this way, the resulting semi-synthetic dataset has realistic covariates *and* we have a full knowledge on how outcomes are generated. In particular, we can restrict to DGPs for which prognostic and predictive covariates are clearly distinct and identifiable. We implement this by selecting *non-overlapping* subsets $\mathcal{I}_{\text{prog}} \subset [d]$ of prognostic and two subsets $\mathcal{I}_0, \mathcal{I}_1 \subset [d]$ of predictive covariates. The prognostic covariates similarly contribute to both POs $Y(0), Y(1)$ through a function $x \mapsto \mu_{\text{prog}}(x_{\mathcal{I}_{\text{prog}}})$, where we let $x_{\mathcal{I}}$ denote the vector $(x_i)_{i \in \mathcal{I}}$. for a set $\mathcal{I} \subset [d]$. The predictive covariates, on the other hand, contribute to either only $Y(0)$ through a function $x \mapsto \mu_{\text{pred0}}(x_{\mathcal{I}_0})$ or to $Y(1)$ through a function $x \mapsto \mu_{\text{pred1}}(x_{\mathcal{I}_1})$.[4] We include a predictive scale $\omega_{\text{pred}} \in \mathbb{R}^+$ that permits to tune the relative strength between the prognostic and predictive contributions to the POs. This full process is detailed in Algorithm 1. All the experiments from Section 5 are produced by varying the inputs of this algorithm; in particular, we consider different types of outcome functions and propensity scores.

---

**Algorithm 1:** Semi-Synthetic Data Generating Process

**Input:** Covariates dataset $\{X^n \in \mathbb{R}^d\}_{n=1}^N$, Prognostic function $\mu_{\text{prog}} : \mathbb{R} \to \mathbb{R}$, Predictive functions $\mu_{\text{pred0}}, \mu_{\text{pred1}} : \mathbb{R} \to \mathbb{R}$, Propensity score $\pi : \mathcal{X} \to \mathbb{R}$, Feature sets size $n_{\mathcal{I}} \in \mathbb{N}^*$, Predictive scale $\omega_{\text{pred}} \in \mathbb{R}^+$, Noise level $\sigma \in \mathbb{R}^+$

**Output:** Semi-synthetic observational dataset $\mathcal{D} = \{(Y^n, X^n, W^n)\}_{n=1}^N$, Prognostic features $\mathcal{I}_{\text{prog}} \subset [d]$, Predictive features $\mathcal{I}_{\text{pred}} \subset [d]$

**Ensure:** $d > 3 \cdot n_{\mathcal{I}}$          /* Avoid overlap between $\mathcal{I}_{\text{prog}}$, $\mathcal{I}_0$ and $\mathcal{I}_1$ */

$\mathcal{I} \leftarrow$ Sample $3 \cdot n_{\mathcal{I}}$ elements from $[d]$ without replacement /* Get relevant features */

$\mathcal{I}_{\text{prog}}, \mathcal{I}_0, \mathcal{I}_1 \leftarrow$ Split $\mathcal{I}$ into 3 sets of size $n_{\mathcal{I}}$    /* Get prog. and pred. features */

$\mathcal{D} \leftarrow \emptyset$                             /* Initialize dataset */

**for** $n \in [N]$ **do**

    $Y(0) \leftarrow \mu_{\text{prog}}(X^n_{\mathcal{I}_{\text{prog}}}) + \omega_{\text{pred}} \cdot \mu_{\text{pred0}}(X^n_{\mathcal{I}_0})$      /* Get untreated outcome */

    $Y(1) \leftarrow \mu_{\text{prog}}(X^n_{\mathcal{I}_{\text{prog}}}) + \omega_{\text{pred}} \cdot \mu_{\text{pred1}}(X^n_{\mathcal{I}_1})$       /* Get treated outcome */

    $W^n \sim \text{Bernoulli}[\pi(X^n)]$          /* Sample treatment assignment */

    $\epsilon \sim \mathcal{N}(0, \sigma)$                        /* Sample noise */

    $Y^n = W^n \cdot Y(1) + (1 - W^n) \cdot Y(0) + \epsilon$      /* Get observed outcome */

    $\mathcal{D} \leftarrow \mathcal{D} \cup \{(Y^n, X^n, W^n)\}$             /* Append dataset */

**end**

**return** $\mathcal{D}$, $\mathcal{I}_{\text{prog}}$, $\mathcal{I}_{\text{pred}} = \mathcal{I}_0 \sqcup \mathcal{I}_1$

---

**Metrics.** After using Algorithm 1, we split the generated dataset into a training set $\mathcal{D}_{\text{train}}$ and a testing set $\mathcal{D}_{\text{test}}$ ($80\% - 20\%$ split). We fit a model $\hat{\tau}$ to estimate the CATE on the training set $\mathcal{D}_{\text{train}}$ and evaluate the model on the held-out test set $\mathcal{D}_{\text{test}}$. As aforementioned, our purpose is to assess if this

---

[3]In fact, due to the DGP used in IHDP *all* important variables are both predictive and prognostic, allowing for no interesting distinctions between discoveries.

[4]Note that the expected difference between the POs can be written as $\mathbb{E}[Y(1) - Y(0) \mid X = x] = \omega_{\text{pred}} \cdot [\mu_{\text{pred1}}(x_{\mathcal{I}_1}) - \mu_{\text{pred0}}(x_{\mathcal{I}_0})]$. In this way, the treatment effect depends only on the covariates indexed by $\mathcal{I}_{\text{pred}} = \mathcal{I}_0 \sqcup \mathcal{I}_1$. This indeed corresponds to the definition of predictive covariates given in Section 3.

model has correctly identified the predictive covariates. With our choice of DGP, we know that those predictive covariates correspond to the indices in $\mathcal{I}_{\mathrm{pred}} = \mathcal{I}_0 \sqcup \mathcal{I}_1$. By recalling that the importance attributed to covariate $x_i$ for a prediction $\hat{\tau}(x)$ increases with the absolute value $|a_i(\hat{\tau}, x)|$, we can compute the average proportion of the attribution *correctly* allocated to the predictive covariates:

$$\mathrm{Attr}_{\mathrm{pred}} = \frac{1}{|\mathcal{D}_{\mathrm{test}}|} \sum_{X \in \mathcal{D}_{\mathrm{test}}} \frac{\sum_{i \in \mathcal{I}_{\mathrm{pred}}} |a_i(\hat{\tau}, X)|}{\sum_{i=1}^{d} |a_i(\hat{\tau}, X)|}. \tag{2}$$

Note that $\mathrm{Attr}_{\mathrm{pred}} \in [0, 1]$, where $\mathrm{Attr}_{\mathrm{pred}} = 0$ corresponds to a model that does not identify any predictive covariate and $\mathrm{Attr}_{\mathrm{pred}} = 1$ corresponds to a model that only identifies predictive covariates. Ideally, predictive covariates should be the most important for a model that estimates the CATE; thus we expect good models to score high with respect to this metric. A similar metric $\mathrm{Attr}_{\mathrm{prog}}$ can be defined analogously to measure the fraction of the feature attribution *incorrectly* allocated to the prognostic covariates by replacing $\mathcal{I}_{\mathrm{pred}} \mapsto \mathcal{I}_{\mathrm{prog}}$ in (2). Ideally, $\mathrm{Attr}_{\mathrm{prog}}$ should be zero if all importance is correctly allocated to predictive variables. Finally, we will sometimes also report the standard PEHE metric, i.e. the RMSE of estimating CATE: $\mathrm{PEHE} = \sqrt{n_{\mathrm{test}}^{-1} \sum_{X \in \mathcal{D}_{\mathrm{test}}} [\hat{\tau}(X) - \tau(X)]^2}$.

## 5 Experiments

In this section, we benchmark different types of CATE estimators on their ability to identify predictive covariates through feature importance scores. We study 3 different characteristics of the data generating process which we expect to impact this ability: (i) the relative strength between the prognostic contribution $\mu_{\mathrm{prog}}$ and the predictive contributions $\mu_{\mathrm{pred0}}, \mu_{\mathrm{pred1}}$ to the POs $Y(0), Y(1)$ (Sec. 5.1); (ii) The presence of nonlinearities in the prognostic and predictive functions $\mu_{\mathrm{prog}}, \mu_{\mathrm{pred0}}, \mu_{\mathrm{pred1}}$ (Sec. 5.2), and (iii) the fact that the treatment assignment might be biased according to a nontrivial propensity score $\pi$ (Sec. 5.3). All the experiments are done by varying the inputs of Algorithm 1. The code to reproduce the experiments is available at https://github.com/JonathanCrabbe/ITErpretability and https://github.com/vanderschaarlab/ITErpretability.

**Datasets.** We extract covariates to use in our our benchmarking environment from the following four datasets: TCGA [51], Twins [52], News [53] and ACIC2016 [50], which were selected due to their diverse characteristics in terms of the number of features, mixture of categorical/continuous features and population size. The number of covariates in these datasets range from $d = 39$ to $d = 100$ and we set $n_{\mathcal{I}} = \lfloor 0.2 \cdot d \rfloor$. Refer to Appendix C for details of the datasets.

**Learners.** We consider a number of CATE estimators based on neural networks throughout our experiments. As *direct* estimators, we use [3]'s DR-learner, which relies on a doubly robust pseudo-outcome, and [2]'s X-learner, which uses a weighted average of two direct (singly-robust) treatment effect estimates from both treatment groups. As *indirect* estimators, we use [2]'s T-learner, which fits *two* regression models $\hat{\mu}_w(x)$ (one for each treatment group) and sets $\hat{\tau}(x) = \hat{\mu}_1(x) - \hat{\mu}_0(x)$, and S-learner, which includes the treatment assignment variable $W$ as a standard feature in a *single* regression $\hat{\mu}(x, w)$ and sets $\hat{\tau}(x) = \hat{\mu}(x, 1) - \hat{\mu}(x, 0)$. Finally, we consider [10]'s TARNet which can be seen as a *hybrid* between S- and T-learner [4]: it learns a representation $\Phi(x)$ *shared* between treatment groups, which is used by *treatment-specific* outcome heads $h_w(\Phi(x))$ so that $\hat{\tau}(x) = h_1(\Phi(x)) - h_0(\Phi(x))$. In the experiments with confounding, we also use [10]'s CFRNet, which differs from TARNet only in a regularization term that encourages the representation to be *balanced* (follow a similar distribution) across treatment groups. We discuss all models in more detail in Appendix C, where we also detail how we fix hyperparameters across all models to ensure similar capacity.

### 5.1 Experiment 1: Altering the Strength of Predictive Effects

**Setup.** We begin by investigating how the strength of predictive effects *relative* to prognostic effects influences the ability of different learners to discover predictive covariates. This is an interesting question, as in practice predictive signals are often assumed to be much *weaker* than prognostic ones [2, 13, 16]. Thus, ideally estimators should be able to correctly identify predictive covariates even when effects are weak; yet some of [16]'s empirical results comparing different random forest-based learners across DGPs with different predictive effects, show that this is not always the case. For our experiments, we thus continuously vary predictive effect size in a DGP with a linear parametrization for the prognostic and predictive functions:

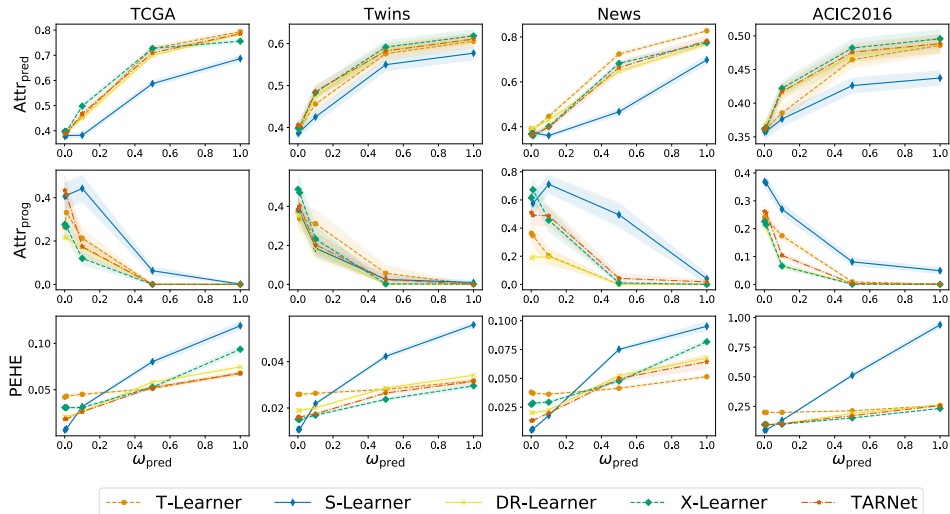

Figure 2: Performance comparison in terms of $\text{Attr}_{\text{pred}}$ (top, higher is better), $\text{Attr}_{\text{prog}}$ (middle, lower is better) and PEHE (bottom, lower is better) when varying the predictive scale, using four feature datasets (TGCA, Twins, News, ACIC). Averaged across multiple runs, shaded areas indicates one standard error.

$\mu_{\text{prog}}(x_{\mathcal{I}_{\text{prog}}}) = \alpha_{\text{prog}}^{\mathsf{T}} x_{\mathcal{I}_{\text{prog}}}$, $\mu_{\text{pred0}}(x_{\mathcal{I}_0}) = \alpha_{\text{pred0}}^{\mathsf{T}} x_{\mathcal{I}_0}$ and $\mu_{\text{pred1}}(x_{\mathcal{I}_1}) = \alpha_{\text{pred1}}^{\mathsf{T}} x_{\mathcal{I}_1}$ with weights sampled randomly $\alpha_{\text{prog}}$, $\alpha_{\text{pred0}}$, $\alpha_{\text{pred1}} \sim U([-1,1]^{n_{\mathcal{I}}})$, where $U$ denotes the uniform distribution. To vary the relative strength between the prognostic and the predictive contributions to the POs, we change the predictive scale $\omega_{\text{pred}} \in \{10^{-3}, 10^{-2}, 10^{-1}, 0.5, 1\}$. Here, treatments are assigned completely at random, i.e. $\pi(x) = 0.5$.

**Results.** In Fig. 2 we present results on correct attributions $\text{Attr}_{\text{pred}}$ and misattributions $\text{Attr}_{\text{prog}}$, as well as the standard PEHE metric for comparison, for all datasets. We make a number of interesting observations: ① *Attribution trends indeed vary with $\omega_{\text{pred}}$*. Correct predictive attributions $\text{Attr}_{\text{pred}}$ substantially increase as $\omega_{\text{pred}}$ increases for all learners and across all datasets considered – this confirms the intuition that the stronger the predictive effects are, the easier it is to identify their origin. Conversely, prognostic misattributions $\text{Attr}_{\text{prog}}$ decrease as $\omega_{\text{pred}}$ increases, indicating that one reason for the low $\text{Attr}_{\text{prog}}$ at low $\omega_{\text{pred}}$ is that learners *confuse* prognostic effects for predictive ones. ② *Comparing learners, S-Learners appear to struggle most to make correct attributions*. The most salient observation across all datasets is that the S-Learner does does substantially worse at $\text{Attr}_{\text{pred}}$. With the exception of the Twins dataset, this usually also translates into higher $\text{Attr}_{\text{prog}}$ than all other learners. We believe that this is because the S-Learner uniquely neither has a treatment-group specific component (like T-learner and TARNet do) nor models CATE directly (like DR- and X-learner do); learned treatment effect heterogeneity thus has to arise through learned interactions with the treatment indicator – which appears to lead to less reliable predictive covariate discoveries. All other methods perform very similar to each other. ③ *Using attribution metrics indeed leads to interesting new insights relative to considering only PEHE*. We observe that the S-learner does *best* in terms of PEHE when predictive strength is low, while the T-learner does worst (as discussed below, this is in line with expectations and empirical observations previously made in e.g. [2, 13]) – yet, as we saw above, this does not translate into better discoveries. Similarly, the better performance in terms of PEHE of some other strategies relative to T-learner when $\omega_{\text{pred}}$ is small also does not lead to better discoveries. We attribute this to the fact that when $\omega_{\text{pred}}$ is very small, PEHE will favour any method that outputs near-zero treatment effects; indeed all considered methods except the T-learner incorporate an implicit inductive bias that shrinks effects [13] – which appears to help only in terms of PEHE. Note also that PEHE is not directly comparable across different values of $\omega_{\text{pred}}$ as it naturally increases as the scale of CATE changes. Finally, we investigate performance on this task using a dataset with higher dimensional input in Appendix C.4.

## 5.2 Experiment 2: Incorporating Nonlinearities

**Description.** In practice, there is no particular reason to expect that POs are linear functions of the covariates. Next we therefore investigate how nonlinearities in the POs influence the ability of CATE

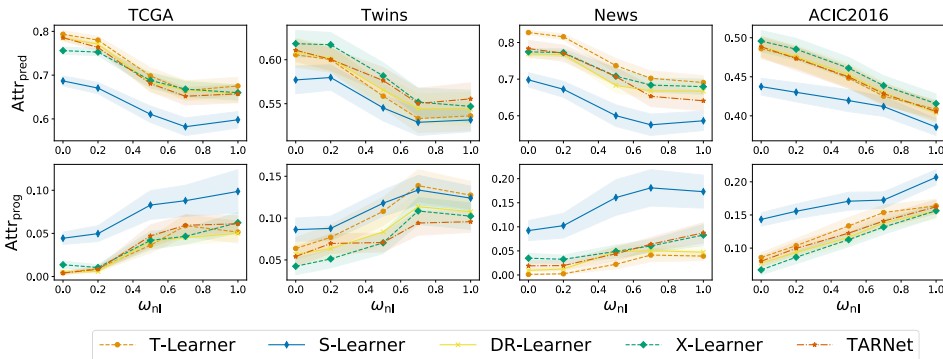

Figure 3: Performance comparison in terms of $\text{Attr}_{\text{pred}}$ (top, higher is better) and $\text{Attr}_{\text{prog}}$ (bottom, lower is better) when varying the nonlinearity scale, using four feature datasets (TGCA, Twins, News, ACIC). Averaged across multiple runs, shaded areas indicates one standard error.

estimators to identify predictive covariates. To do so, we use a parametrization for the prognostic and predictive functions that allows us to control the strength of the nonlinearities through parameter $\omega_{\text{nl}}$: $\mu_{\text{prog}}(x_{\mathcal{I}_{\text{prog}}}) = (1 - \omega_{\text{nl}}) \cdot \alpha_{\text{prog}}^{\mathsf{T}} x_{\mathcal{I}_{\text{prog}}} + \omega_{\text{nl}} \cdot \chi(\alpha_{\text{prog}}^{\mathsf{T}} x_{\mathcal{I}_{\text{prog}}})$, $\mu_{\text{pred0}}(x_{\mathcal{I}_0}) = (1 - \omega_{\text{nl}}) \cdot \alpha_{\text{pred0}}^{\mathsf{T}} x_{\mathcal{I}_0} + \omega_{\text{nl}} \cdot \chi(\alpha_{\text{pred0}}^{\mathsf{T}} x_{\mathcal{I}_0})$ and $\mu_{\text{pred1}}(x_{\mathcal{I}_1}) = (1 - \omega_{\text{nl}}) \cdot \alpha_{\text{pred1}}^{\mathsf{T}} x_{\mathcal{I}_1} + \omega_{\text{nl}} \cdot \chi(\alpha_{\text{pred1}}^{\mathsf{T}} x_{\mathcal{I}_1})$ with weights and nonlinearity sampled randomly $\alpha_{\text{prog}}, \alpha_{\text{pred0}}, \alpha_{\text{pred1}} \sim U([-1,1]^{n_{\mathcal{I}}})$, $\chi \sim U(\mathcal{F})$, where $\mathcal{F}$ is a set of 10 nonlinear functions $\mathbb{R} \to \mathbb{R}$ (specified in Appendix C.5). To vary the strength of the nonlinearities, we let the nonlinearity scale vary $\omega_{\text{nl}} \in [0, 1]$. We note that $\omega_{\text{nl}} = 0$ corresponds to linear function as in the previous experiment. On the other hand, $\omega_{\text{nl}} = 1$ corresponds to purely nonlinear functions. Here, we set $\omega_{\text{pred}} = 1$ – a setting for which all learners performed well in the previous experiment and $\pi(x) = 0.5$.

**Results.** We present attribution score results across all datasets in Fig. 3. We find that ① *Attribution trends vary with $\omega_{\text{nl}}$.* As $\omega_{\text{nl}}$ increases and the underlying DGP is dominated by the nonlinearity, hence becoming more difficult to learn, we observe that correct attribution ($\text{Attr}_{\text{pred}}$) decreases for all methods. Also here, we observe that this is mirrored by an *increase* in confusion of prognostic effects for predictive ones (as seen in the increasing $\text{Attr}_{\text{prog}}$). Further, we note that ② *Relative ordering of methods does not change.* The S-Learner continues to underperform compared to all other learners and the performance gap does not substantially change across values of $\omega_{\text{nl}}$.

## 5.3 Experiment 3: The Effect of Confounding

**Description.** Finally, we turn to examine the effect of *confounding*, i.e. covariate shift between treatment groups resulting from treatment assignment being based on observables – a problem that much of the ML literature proposing CATE estimators has focussed on (e.g. [9, 10, 12]). A popular solution to deal with said covariate shift has been to rely on *balancing* regularization that penalizes distributional distance (here: $\text{MMD}^2$) between treatment groups in representation space; here we therefore also consider [10]'s CFRNet which is identical to TARNet but includes a discrepancy-based regularization term controlled by hyperparameter $\gamma$. We note that if the covariates that determine treatment assignment are either predictive or prognostic – i.e. they are true *confounders* – it is generally not possible to remove *all* covariate shift without removing predictive/prognostic information. We therefore consider a final experiment where we structurally vary not only the degree of assignment bias (through propensity scale $\omega_{\pi}$) but also what *type of information* assignment is based on. We achieve this by modifying the propensity score: $\pi(x) = \texttt{Sigmoid}(\omega_{\pi} \cdot Z_{\text{score}}[\psi(x)])$, where $Z_{\text{score}}[\cdot]$ indicates normalization across the generated training dataset (this ensures well-behaved propensity scores centered at 0.5) and $\psi : \mathcal{X} \to \mathbb{R}$ controls the type of confounding.

We consider 3 types of confounding, each corresponding to a different choice for $\psi$. *Predictive confounding* corresponds to setting $\psi = \mu_{\text{pred1}} - \mu_{\text{pred0}}$. It mimics a scenario where treatment assignment is biased towards those with characteristics making them most likely to respond well to it, e.g. a doctor assigning treatment with knowledge of CATE. *Prognostic confounding* corresponds to setting $\psi = \mu_{\text{prog}}$. It mimics the most classical confounding setting where treatment assignment is biased towards those with characteristics making them more likely to have a good outcome regardless of treatment, e.g. self-selection into a treatment program. Finally, we consider a *non-confounded*

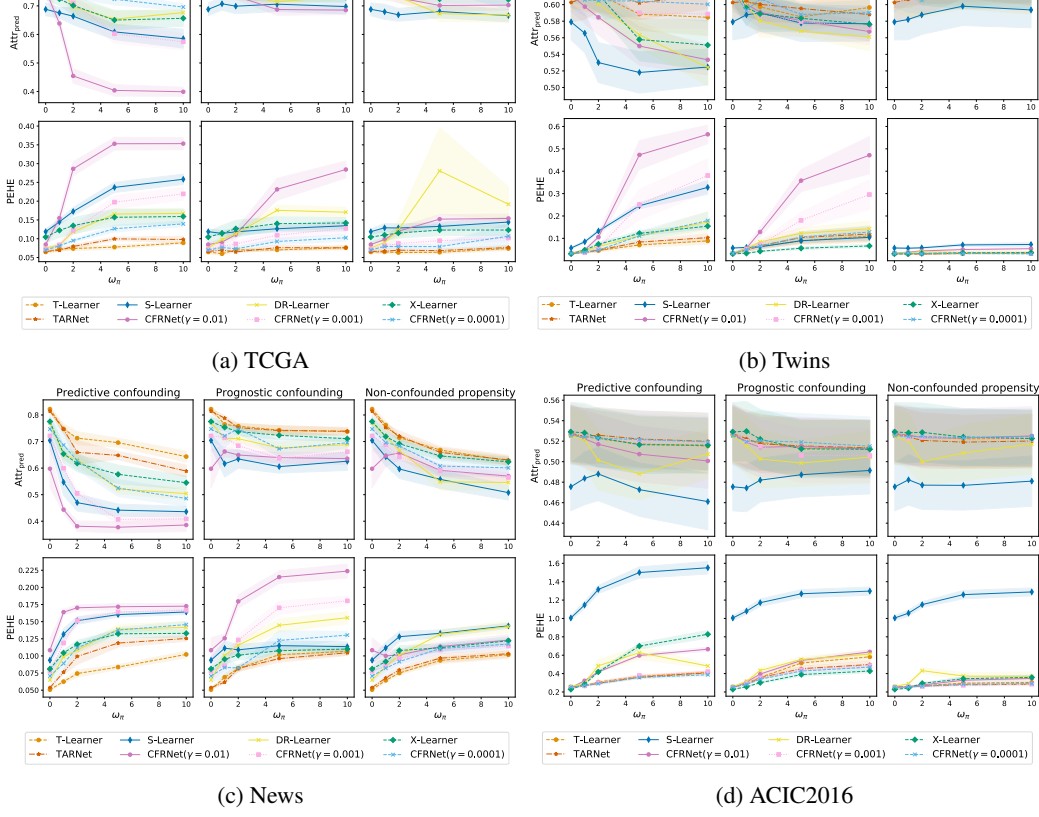

(a) TCGA

(b) Twins

(c) News

(d) ACIC2016

Figure 4: Performance comparison when increasing the propensity scale. Averaged across multiple runs, shaded areas indicates one standard error.

*propensity*: $\psi(x) = x_i$ for some irrelevant covariate $i \notin \mathcal{I}_{\mathrm{prog}} \sqcup \mathcal{I}_{\mathrm{pred}}$. In this case, the treatment selection is not based on a covariate that affects outcome; note that the distribution of covariates in $\mathcal{I}_{\mathrm{pred}} \sqcup \mathcal{I}_{\mathrm{prog}}$ might still differ across treatment groups if they are correlated with the chosen 'irrelevant'. Note that for $\omega_\pi = 0$, all settings are identical and reduce to the previous $\pi(x) = 0.5$. For the potential outcomes, we consider the previous setting with $\omega_{\mathrm{pred}} = 1$ and $\omega_{\mathrm{nl}} = 0$. We believe that such an explicit distinction between different confounding types has not yet been investigated in related work.

**Results.** We present results comparing $\mathrm{Attr}_{\mathrm{pred}}$ and PEHE across the three settings for each dataset in Fig. 4. We make numerous interesting observations: ① *Attribution trends indeed vary with $\omega_\pi$.* Also in this experiment attribution scores vary systematically as we change $\omega_\pi$; across the different settings the quality of attributions worsens the more biased treatment assignment is, albeit only marginally for some settings. ② *There are systematic differences across the three settings.* We observe that performance deteriorates the most in the setting with predictive confounding. We believe that this is a result of less observed variation across predictive covariates *within* a group due to the assignment bias, making it harder for any model to learn that treatment effect varies systematically across these covariates. Overall, we observe the least deterioration in attribution performance in the prognostic confounding setting. That is because, while the prognostic confounding and non-confounded setting are almost identical in terms of $\mathrm{Attr}_{\mathrm{pred}}$ for TCGA, Twins and ACIC, we observe an interesting difference in News: Perhaps surprisingly, we find that performance as measured by $\mathrm{Attr}_{\mathrm{pred}}$ in News deteriorates *less* in the prognostic confounding setting than in the non-confounded setting. We attribute this to two *competing* forces being at play here: First, having any sort of covariate imbalance, as in the non-confounded setting, appears to have the potential to lead to a decrease in $\mathrm{Attr}_{\mathrm{pred}}$. Second, in the prognostic confounding setting, there is a similar effect as in the predictive setting that can offset some of the initial performance decrease: as prognostic covariates now have less variation within a group, models are less likely to *misattribute* a predictive effect to them. ③ *CFRNet's balancing regularization has different effects across settings.* We find that

the addition of the balancing regularization term can lead to a very large drop in $\text{Attr}_{\text{pred}}$ in the predictive confounding setting – this is expected as overly aligning distributions should lead to a loss in explanatory power, which is also reflected in the PEHE. In the other two settings, no such effect is visible when considering $\text{Attr}_{\text{pred}}$; yet, we observe that PEHE does worsen considerably in the prognostic confounding setting. We believe that this is an effect of the prognostic component of the POs being estimated less accurately due to the balancing regularization, making the CATE estimate, the difference between the two estimated POs, less accurate overall.

# 6   Discussion

In this paper, we have introduced the ITErpretability benchmark, a new environment to benchmark CATE models with the help of feature importance methods. We empirically demonstrated on various datasets that this benchmark provides insights that are *not* accessible with the metrics and benchmarks considered standard in the CATE literature. We believe that this work opens up many interesting avenues for future research: First, our environment could be used to extend insights to many more of the CATE estimators proposed in recent literature. One could, for example, replicate our propensity experiments with learners tackling confounding with methods other than balancing, e.g. importance weighting as in [12], or compare performance across different classes of underlying ML methods – i.e. compare how discoveries differ across implementations relying on neural networks with e.g. random forests as in [16] or the gaussian processes of [7, 8]. Second, it may be worthwhile to further study possible failure modes of the considered learning strategies within and beyond our benchmarking environment. This could be achieved both by considering an even wider range of DGPs to investigate how our results generalize empirically, and by complementing our empirical findings with theoretical ones studying under which conditions what types of learners are *guaranteed* to discover the correct predictive covariates. Third, we believe that it would be interesting to consider how to *improve existing* or *develop new* CATE estimation strategies with the help of interpretability techniques or insights derived from experiments such as the ones from Sec. 5. Finally, note that we have exclusively focused on *feature importance* methods here. Another possible extension of this work would then be to perform a similar study with other type of explanation methods, such as example-based explanation methods like Influence Functions [55] and hybrid methods like SimplEx [56].

## Acknowledgments and Disclosure of Funding

The authors are grateful to Javier Abad Martinez and Bogdan Cebere for implementing and running preliminary, yet different, experiments in the early and explorative stages of the project. Jonathan Crabbé is funded by Aviva, Alicia Curth is funded by AstraZeneca, Ioana Bica is funded by the Alan Turing Institute (under the EPSRC grant EP/N510129/1) and Mihaela van der Schaar is funded by the Office of Naval Research (ONR under the grant NSF 1722516). The authors have no competing interests to disclose.

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
