# Appendix

This Appendix is structured as follows: in Section A, we give additional background information on CATE estimation. In Section B, we discuss feature importance methods in more detail. Finally, in Section C we give further details on experiments and present additional results.

## A    Further Background on CATE Estimation

In this section, we discuss the unique characteristics of CATE estimation in more detail. As outlined in section 2, we consider three characteristics most important:

• **1. The need to rely on untestable assumptions.**  To infer causal effects from observational data, one needs to make strong *untestable* assumptions which ensure *identifiability* of a treatment effect and should be assessed by a domain expert in practice. Such assumptions are used to assure that treated and untreated individuals are exchangeable, so that $\mathbb{E}_P[Y(w) \mid X = x] = \mathbb{E}_P[Y \mid W = w, X = x]$. As is standard in related literature, we rely on the strong ignorability conditions [35] giving rise to

**Assumption 1.** [Consistency, ignorability, and positivity] *Consistency*: If individual $n$ is assigned treatment $w^n$, we observe the associated potential outcome $Y^n = Y^n(w^n)$. *Ignorability*: there are no hidden confounders, such that $Y(0), Y(1) \perp W \mid X$. *Positivity*: treatment assignment is non-deterministic, i.e. $0 < \pi(x) < 1, \forall x \in \mathcal{X}$.

• **2. The presence of covariate shift due to confounding.** Even when ignorability holds because $X$ contains all confounders, a non-constant propensity score $\pi(x)$ will lead to covariate shift between the two treatment groups. When treatment effects are estimated indirectly by first obtaining estimates of $\mu_w(x)$, this can be problematic during empirical risk minimization as the observed population distribution then does not correspond to the target (marginal) distribution of characteristics.

• **3.  The target label of interest is absent.**    In fully randomized experiments, identifying assumptions hold by construction and the distribution of covariates across treatment arms is identical (in expectation) – yet CATE estimation remains non-trivial. This is because the true target label $Y(1) - Y(0)$ is absent even in experimental studies. Because $Y(1)$ and $Y(0)$ are available separately, outcome regressions estimating $\mu_{w(x)} = \mathbb{E}_P[Y \mid W = w, X = x]$ can be performed and $\hat{\tau}(x)$ can be estimated indirectly as $\hat{\mu}_1(x)$ and $\hat{\mu}_0(x)$. Nonetheless, as we discuss in Section 2 it is also possible to target $\tau(x)$ directly; the two estimation strategies have different theoretical strengths [4, 13].

## B    Feature Importance Methods

### B.1    Useful Properties for CATE Interpretability

With the specificity of the CATE setting in mind, let us describe some desirable properties of the importance scores $a_i, i \in [d]$. The presence (or absence) of these properties in popular feature importance methods is summarized in Table 1.

**Sensitivity.** The covariates that do not affect the CATE model are given zero contribution. More formally, if for some $i \in [d]$ we have $\hat{\tau}(x) = \hat{\tau}(x_{-i})$ for all $x \in \mathcal{X}$ , then $a_i(\hat{\tau}, x) = 0$ for all $x \in \mathcal{X}$. This allows to discard covariates that are irrelevant for the CATE model.

**Completeness.** Summing the importance scores gives the shift between the CATE and a baseline. More formally, for all $x \in \mathcal{X}$, we have:

$$\sum_{i=1}^{d} a_i(\hat{\tau}, x) = \hat{\tau}(x) - b,$$

where $b \in \mathbb{R}$ is a constant baseline. In this way, each importance score $a_i$ can be interpreted as the contribution from covariate $i$ of $x$ to have a CATE that differs from the baseline $b$. Note that the choice of the baseline differs from one method to another. For instance, the baseline for SHAP and Expected Gradient [41] is the average treatment effect: $b = \mathbb{E}_{X \sim P}[\hat{\tau}(X)]$. Alternatively, the baseline employed in Integrated Gradient and DeepLift is the treatment effect for a baseline patient with covariates $\bar{x}$: $b = \hat{\tau}(\bar{x})$. Finally, Lime uses a zero baseline: $b = 0$.

**Linearity.** The importance score is linear with respect to the black-box function $f$. More formally, this means that for all covariate $i \in [d]$, real numbers $\alpha, \beta \in \mathbb{R}$, functions $f_a, f_b : \mathcal{X} \to \mathcal{Y}$ we have $a_i(\alpha \cdot f_a + \beta \cdot f_b, \cdot) = \alpha \cdot a_i(f_a, \cdot) + \beta \cdot a_i(f_b, \cdot)$. If the CATE model $f = \hat{\tau}$ is written directly in terms of the estimated potential outcomes $\hat{\tau} = \hat{\mu}_1 - \hat{\mu}_0$, this allows to write:

$$a_i(\hat{\tau}, x) = a_i(\hat{\mu}_1, x) - a_i(\hat{\mu}_0, x).$$

This formulation renders the distinction between prognostic and predictive covariates intuitive. If $x_i$ is a prognostic covariate, one expects $a_i(\hat{\mu}_1, x) = a_i(\hat{\mu}_0, x)$ so that $a_i(\hat{\tau}, x) = 0$, which implies that $i$ is not relevant to explain effect heterogeneity. On the other hand, if $x_i$ is a predictive covariate, one expects $a_i(\hat{\mu}_1, x) \neq a_i(\hat{\mu}_0, x)$ so that $a_i(\hat{\tau}, x) \neq 0$, which implies that $x_i$ is relevant to explain effect heterogeneity.

**Model Agnosticism.** The feature importance score can be computed for all CATE model $\hat{\tau} : \mathcal{X} \to \mathcal{Y}$. Some methods only work with a restricted family of models, which prevents them from being model agnostic. A typical example is Integrated Gradient, which requires the model $\hat{\tau}$ to be differentiable with respect to its input. Another example is DeepLift that requires $\hat{\tau}$ to be represented by a deep-neural network.

**Implementation Invariance.** The feature attribution would be the same for two functionally equivalent models. This means that if we have two CATE models $\hat{\tau}_1$ and $\hat{\tau}_2$ such that $\hat{\tau}_1(x) = \hat{\tau}_2(x)$ for all $x \in \mathcal{X}$, this implies that $a_i(\hat{\tau}_1, x) = a_i(\hat{\tau}_2, x)$ for all $x \in \mathcal{X}$ and all $i \in [d]$. While this property might seem trivial, we note that it is not fulfilled when the attribution methods explicitly depend on the model's architecture. An example of such methods are the ones that use neuron activations, like DeepLift and LRP.

Table 1: Properties of popular feature importance methods. Note that Shap also has a gradient-based implementation called GradientShap. Hence, it also belongs to the Gradient-Based category.

| Type | Name | Sensitivity | Completeness | Linearity | Impl. Invariance |
|---|---|---|---|---|---|
| Gradient-Based | Saliency [38] | ✓ | ✗ | ✓ | ✓ |
| | Integrated Gradients [39] | ✓ | ✓ | ✓ | ✓ |
| Perturbation-Based | Lime [42] | ✗ | ✓ | ✗ | ✓ |
| | Feature Ablation | ✓ | ✗ | ✗ | ✓ |
| | Feature Permutation | ✓ | ✗ | ✗ | ✓ |
| | Shap [40] | ✓ | ✓ | ✓ | ✓ |
| Neuron Activation | LRP [45] | ✓ | ✓ | ✓ | ✗ |
| | DeepLift [46] | ✓ | ✓ | ✓ | ✗ |

We note that two methods stand-out in the previous analysis: Integrated Gradients and Shap. The former is much more efficient computationally as it typically requires 50 backwards pass on the model per instance. Shap's complexity, on the other hand, scales exponentially with the number of features $d$ [40]. For datasets with high $d$ (like TCGA), computing Shap for thousands of examples quickly become prohibitively expensive. For this reason, we chose Integrated Gradients as our main explanation method.

## B.2 Quantitative Comparison between Feature Importance Methods

We shall now reproduce the experiments form Sections 5.1 & 5.2 with different feature importance methods. To approximate Shapley values, we use the Monte-Carlo sampling from [57]. We found this approach more computationally efficient than KernelShap.

**Prognostic Scale.** The experiments from Section 5.1 with various feature importance methods is reported in Figure 5. We clearly see that the results for Shap and Integrated Gradients are nearly identical. The predictive accuracy obtained with Feature Ablation is marginally lower, while Feature Permutation substantially underperforms. We note that all the conclusions discussed in Section 5.1 still hold if we replace Integrated Gradients by Shap or Feature Ablation. In particular, the relative ordering between learners is not affected by this choice.

**Nonlinearity Sensitivity.** The experiments from Section 5.2 with various feature importance methods is reported in Figure 6. Again, Shap and Integrated Gradients are closely followed by Feature Ablation

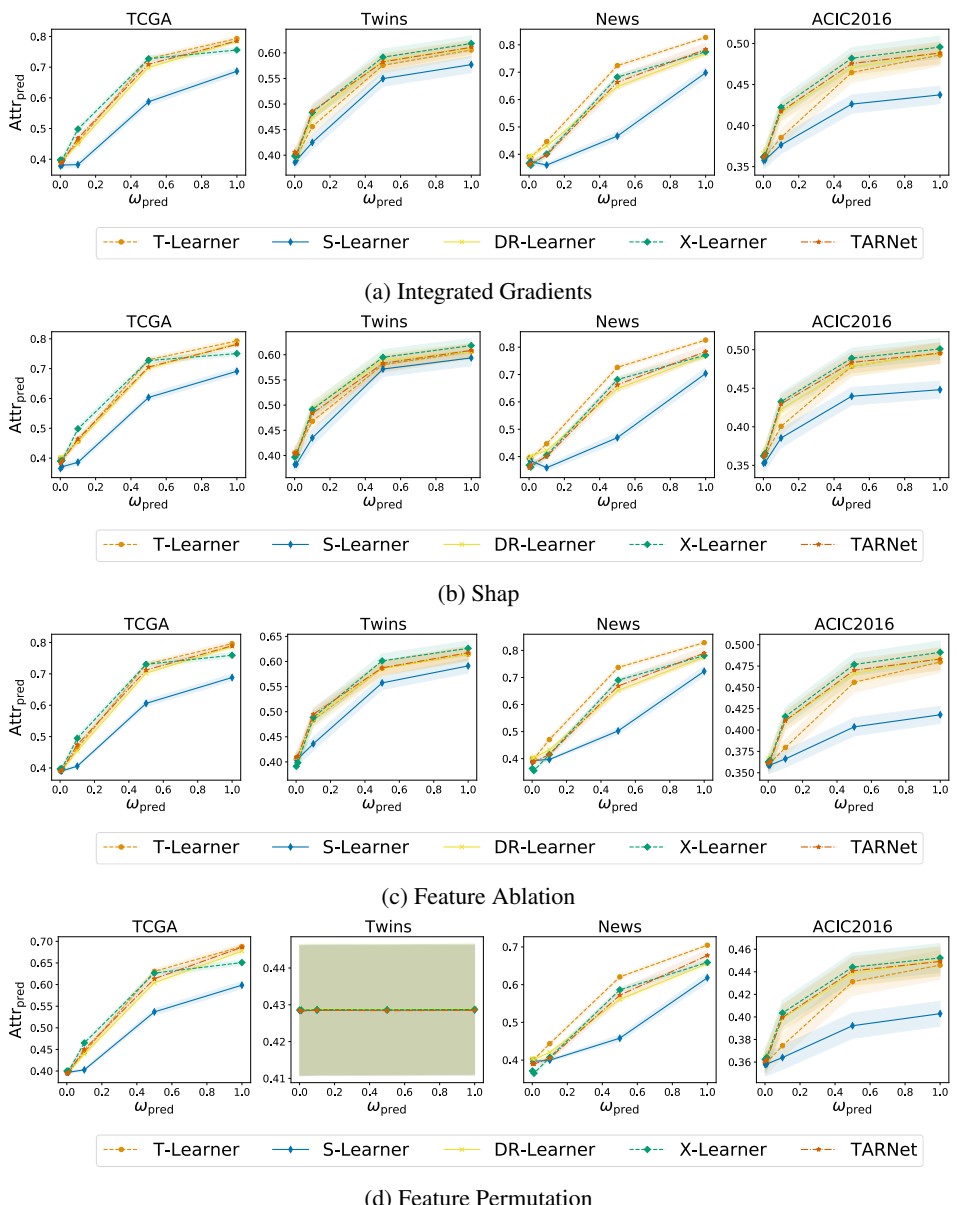

Figure 5: Performance comparison of feature importance methods in terms of $\text{Attr}_{\text{pred}}$ when varying the predictive scale, using four feature datasets (TGCA, Twins, News, ACIC). Averaged across multiple runs, shaded areas indicates one standard error.

and Feature Permutation significantly underperforms. All the conclusions discussed in Section 5.2 still hold if we replace Integrated Gradients by Shap or Feature Ablation. In particular, the relative ordering between learners is not affected by this choice.

# C  Experimental Details and Additional Results

## C.1  CATE Model and Implementation Details

We use two direct learners, both of which use a first-stage regression step to estimate nuisance parameters $\eta = (\mu_0, \mu_1, \pi)$ and then use these to create a surrogate for the treatment effect in the

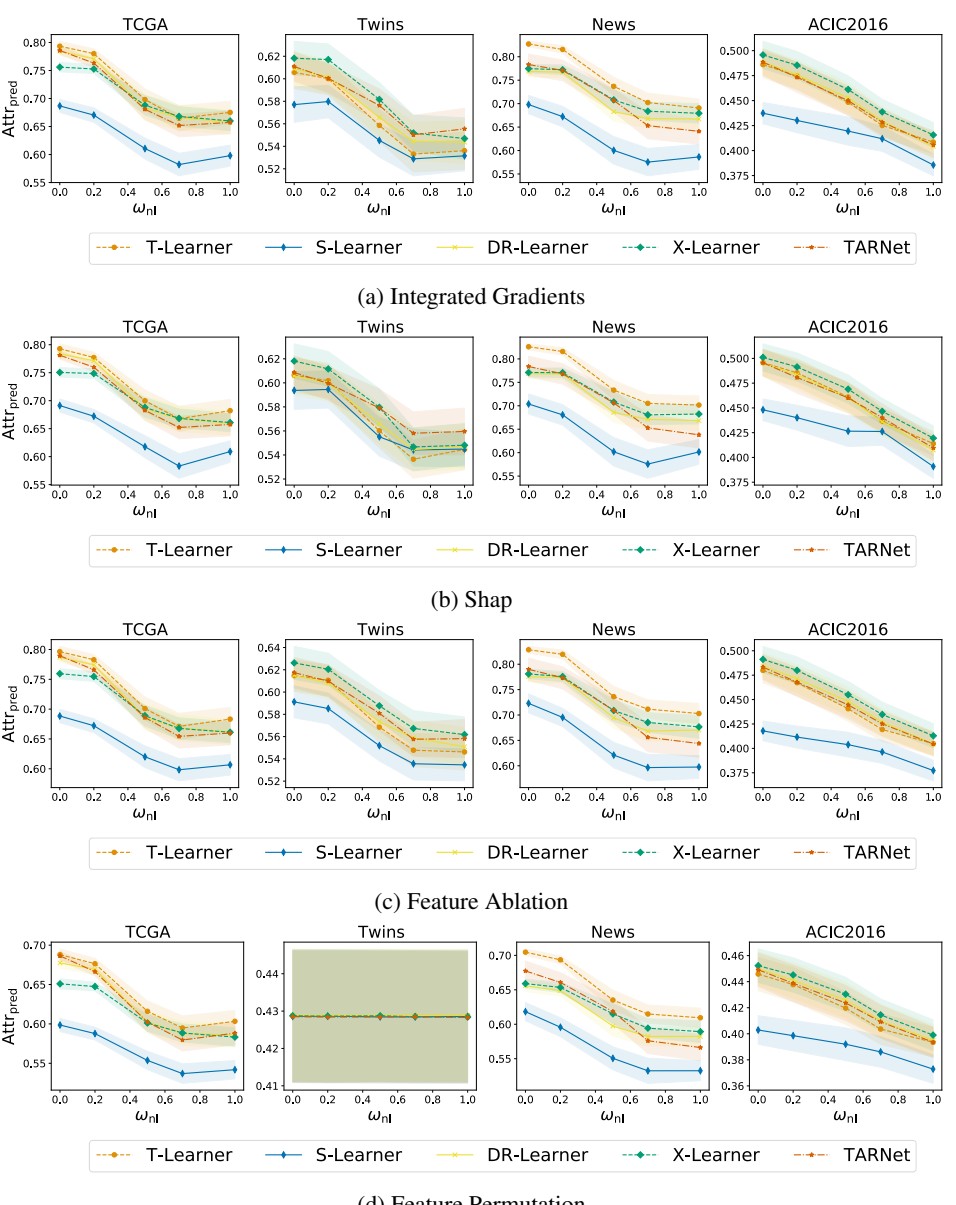

Figure 6: Performance comparison of feature importance methods in terms of $\mathrm{Attr}_{\mathrm{pred}}$ when varying the nonlinearity scale, using four feature datasets (TGCA, Twins, News, ACIC). Averaged across multiple runs, shaded areas indicates one standard error.

second stage. In particular, [3]'s DR-learner uses the pseudo-outcome

$$\tilde{Y}_{DR,\hat{\eta}} = \left( \frac{W}{\hat{\pi}(X)} - \frac{(1-W)}{1-\hat{\pi}(X)} \right) Y + \left[ \left( 1 - \frac{W}{\hat{\pi}(X)} \right) \hat{\mu}_1(x) - \left( 1 - \frac{1-W}{1-\hat{\pi}(X)} \right) \hat{\mu}_0(X) \right] \quad (3)$$

which is based on the doubly-robust AIPW estimator [58] and is unbiased if either propensity score *or* outcome regressions are correctly specified. [2]'s X-learner, on the other hand, creates *two* pseudo-outcomes, one to be used in each treatment group: $\hat{\tau}_1$ is estimated by using pseudo outcome $Y - \hat{\mu}_0(X)$ in a regression using only treated individuals, while $\hat{\tau}_0$ is estimated by using pseudo outcome $\hat{\mu}_0(X) - Y$ in a regression using only control individuals. The two estimators are then combined using $\hat{\tau}(x) = g(x)\hat{\tau}_1(x) + (1 - g(x))\hat{\tau}_0(x)$ for some weighting function $g(x)$; as proposed by [2] we rely on $g(x) = \pi(x)$. Other direct learners, which we do not include in our experiments as they rely on similar principles, have also been studied in the literature: e.g. the RA- and PW-learner in [4] as singly-robust versions of the DR-learner, or the doubly-robust R-learning strategy [59, 60] which uses [61]'s approach for semiparametric regression through orthogonalization with respect to the nuisance functions $\pi(x)$ and $\mu(x) = \mathbb{E}[Y|X = x]$ (the unconditional outcome expectation), instead of the AIPW-transformation.

Indirect learners (T-learner, S-learner and TARNet) are as described in the main text.

We use the PyTorch implementations of all models provided in the python package `Catenets`[5]; as [4] we ensure that each estimated function ($\hat{\mu}_w(x)$, $\hat{\pi}(x)$ and $\hat{\tau}(x)$) has access to the same amount of hidden layers and units in total (2 hidden layers with 100 units and a final prediction layer; for TARNet/CFRNet this means that the representation $\Phi$ and the outcome heads $h_w$ each have 1 hidden layer) and each architecture can hence represent similarly complex nuisance functions. We use dense layers with ReLU activation function. All models are trained using the Adam optimizer with learning rate $10^{-4}$, batch size 1024 and early stopping on the validation set (which represents 30% of the initial training set).

We used a virtual machine with 6 CPUs, an Nvidia K80 Tesla GPU and 56GB of RAM to run all experiments.

## C.2 Dataset Details

**TCGA.** The TCGA dataset [51] consists of information about gene expression measurements from 9659 cancer patients. We use the same version of the TCGA dataset as in [62][6]. In our experiments, we use as patient covariates the measurements from the 100 most variable genes. These are all continuous features. The data is log-normalized and each feature is scaled in the $[0, 1]$ interval.

**Twins.** The Twins dataset [52] consists of information from 11400 twin births in the USA recorded between 1989-1991. Each twin pair is characterized by 39 covariates related to the parents, pregnancy and birth; these represent a mixture of continuous and categorical features. In our experiments, the publicly available version of the dataset from `Catenets` is used where we randomly sample one of the twins to observe.

**News.** The News dataset consists of 10000 news items (randomly sampled), each characterized by 2858 word counts [53, 9, 62]. Similarly to [9] we perform Principal Component Analysis and use as covariates for each news item the first 100 principal components (continuous features). We use the same version of the News dataset as used in [62].

**ACIC2016.** The ACIC2016 dataset consists of data from the Collaborative Perinatal Project provided as part of the Atlantic Causal Inference Competition (ACIC2016) [50]. We use the publicly available version of the dataset from the `Catenets` package which consist of 55 covariates (mixture of continuous and categorical ones) for 2200 patients. Note that the same version of the dataset was used in [13].

For information about how each dataset was collected and curated, refer to the corresponding references. Note that all of these datasets are publicly available. Each dataset undergoes a 80%/20% split for training/testing respectively. Moreover 30% of the training dataset is used for validation as part of the early stopping procedure performed by the `Catenets` package. The feature importance metrics are computed for up to 1000 examples from the test set, while the PEHE is computed over

---

[5] https://github.com/AliciaCurth/CATENets
[6] https://github.com/d909b/drnet

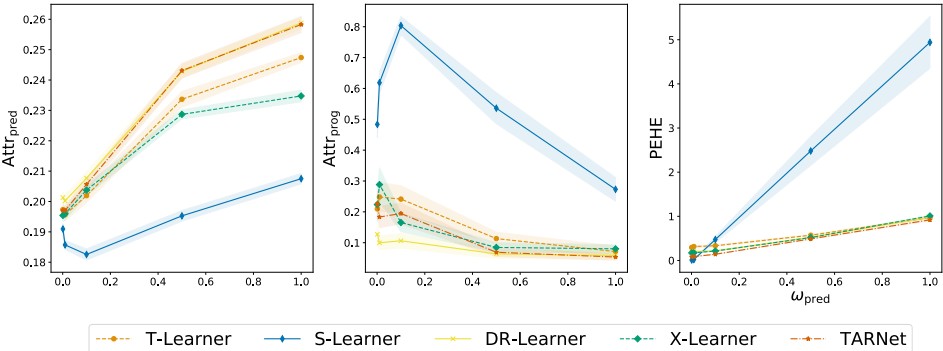

Figure 7: Performance comparison in terms of $\mathrm{Attr}_{\mathrm{pred}}$ (left, higher is better), $\mathrm{Attr}_{\mathrm{prog}}$ (middle, lower is better) and PEHE (right, lower is better) when varying the predictive scale, using a high-dimensional feature space (5000 genes) for the TCGA dataset. Averaged across multiple runs, shaded areas indicates one standard error.

the entire test set. The results are averaged over 30 random seeds for Experiments 1 & 2 and over 10 random seeds for Experiment 3 (note that there are three experimental settings here for the different types of confounding).

### C.3 Feature Importance Methods Implementation

We use the Pytorch implementation of all feature importance methods provided in the Python package Captum[7]. For the relevant feature importance methods, we set the zero vector as a baseline input: $\bar{x} = 0$. We see mainly two limitations for Integrated Gradients, both of which can be addressed in our setting.

1. To compute Integrated Gradients, the black-box needs to be differentiable with respect to its input features. In our setting this is not a problem since all learners are parametrized with feed-forward neural networks.

2. Integrated Gradients is sensitive to the choice of the baseline $\bar{x}$, as discussed in [63]. For this reason, it is important to choose a baseline with a clear interpretation. In our setting, the baseline $\bar{x} = 0$ has a natural interpretation since all the covariates we manipulate are either normalized in the interval $[0, 1]$ (this is the case for the TCGA, News and Twins datasets) or standardized (this is the case for ACIC2016). Hence, the baseline corresponds to the minimal features in the former case and to the mean features in the later case.

### C.4 Experiment 1: Altering the Strength of Predictive Effects – Extension to High-Dimensional Feature Spaces

In Fig. 7 we also report performance when using a version of the TCGA dataset with a high-dimensional feature space. In particular, we have obtained a version of the TCGA data which has the $d = 5000$ most variable genes as features (instead of 100 as used in our initial experiments). Out of the 5000 features, we set $n_{\mathcal{I}} = 500$ of them (10%) to be important as prognostic and predictive for each potential outcome. We notice that in this setting, the different learners have poor absolute performance in discovering the predictive features, while relative performance remains similar to the smaller scale experiments. Thus, our benchmark suggests that developing CATE learners that can better identify predictive covariates in a high dimensional setting is needed and would represent an interesting direction for future work.

### C.5 Experiment 2: Incorporating Nonlinearities

Each nonlinear function $\chi$ is sampled randomly from the function set $\mathcal{F} = \{\chi_1 : x \mapsto |x|, \; \chi_2 : x \mapsto \exp(-x^2), \; \chi_3 : x \mapsto (1 + x^2)^{-1}, \; \chi_4 : x \mapsto \cos(x), \; \chi_5 : x \mapsto \sin(x), \; \chi_6 : x \mapsto \arctan(x), \; \chi_7 : x \mapsto \tanh(x), \; \chi_8 : x \mapsto \log(1 + x^2), \; \chi_9 : x \mapsto (1 + x^2)^{1/2}, \; \chi_{10} : x \mapsto \cosh(x)\}$.

---

[7]https://captum.ai/