# OpenReview forum: "Benchmarking Heterogeneous Treatment Effect Models through the Lens of Interpretability"
_NeurIPS.cc/2022/Track/Datasets_and_Benchmarks — NeurIPS 2022 Datasets and Benchmarks _

### Official Review · Reviewer_uKLm · 2022-07-12
**Review for "Benchmarking Heterogeneous Treatment Effect Models through the Lens of Interpretability"**

**Rating:** 7
**Confidence:** 5
**Correctness:** I believe the methods and experiments…
**Clarity:** This paper is well written.

**Strengths:**

1. Important question on how to interpret black-box CATE models, especially highlighting the importance of differentiating predictive and prognostic features in causal inference scenarios.

2. Provide a practical pipeline for benchmarking existing CATE estimators in terms of interpretability.

**Weaknesses:**

It is still inconclusive from the results like which methods essentially perform the best. Several factors may influence the benchmark results significantly:

- the heterogeneity of the four datasets

- the chosen feature importance estimation methods

- the picked base models for meta-learners

- the mechanism that generates the semi-synthetic data

All of them narrow the scope of this benchmark for guidance in real-world practice.

**Additional Feedback:**

If the feature importance estimated for predictive and prognostic features matters for the deployment of CATE methods, what is the best practice to evaluate CATE estimators? When one model gets high $Attr_{pred}$, low $Attr_{prog}$ but high PEHE, or when one model gets low $Attr_{pred}$ but low PEHE?

**Documentation:**

A brief readme for running scrips for reproducing three experiments in this paper is provided. However, no detailed documentation is found.

**Ethics:**

No ethical concern.

**Relation To Prior Work:**

This paper makes a thorough review of existing arts in CATE and model explainability.

**Summary And Contributions:**

This paper investigates the feature importance estimation in conditional average treatment effect (CATE) tasks. Specifically, this paper argues that the prognostic and predictive features should be discriminated in CATE. To this aim, the authors propose a semi-synthetic dataset that provides ground truth PEHE and the identification of prognostic and predictive features. On top of it, experiments are performed to compare the existing meta-learners as well as model-specific methods in CATE on the feature importance allocated to predictive and prognostic features.

---

> ### Author Response · Authors · 2022-08-14
> **Response to Reviewer uKLm**
>
> Thank you for your thoughtful comments and suggestions. We are grateful for your appreciation of our paper! We respond to your major points in turn below.
>
> ### Conclusiveness of our results
>
> Allow us to emphasize that the goal of this paper was not to find a method that _always_ performs best – in fact, analogously to ‘No Free Lunch’ Theorems elsewhere in ML, we believe that such a method may not even exist. Instead, our goal was to investigate a new dimension to evaluate/benchmark existing CATE estimators on, and gain some first insights into relative performance of some learners across different scenarios.
>
> Here, we focussed on semi-synthetic DGPs that enabled the simulation of datasets _with characteristics_ that can arise in realistic scenarios. In particular, we investigated the effect of the magnitude of predictive effects, of the degree of nonlinearity (using a wide range of non-linear outcome functions) and of different types of treatment selection bias (predictive confounding, prognostic confounding and non-confounded propensity). Because it will depend on the application at hand what kind of simulation scenario is closest to real-world practice, our investigation allowed to study in more generality how performance differs across _a range of possible scenarios_, which can in turn guide a practitioner’s model choice when specific characteristics of the applications are known (e.g. the likely type of confounding or strength of predictive effect). Note also that, because our benchmark environment is modular, anyone with a specific application in mind could easily replicate our analysis using a different DGP, if that was more applicable to their setting of interest.
>
> We also agree that it would be interesting to further investigate differences across base-models for the learners under investigation; as we highlight in our discussion section, we hope that our benchmark environment could be used for precisely that purpose in future work. Finally, we do investigate the use of different feature importance methods in Appendix B.2, and find that some feature importance methods indeed lead to worse absolute attribution scores, while the relative ordering of CATE learners remains nearly constant across explainers.
>
> ### Code documentation
>
> Thank you very much for your comment! We have now updated the documentation for the code to specify how to add a new dataset or a new learner for evaluation.
>
> We would like to highlight that our code has been implemented to easily allows for new datasets or other CATE learners to be used for evaluation. In particular, to add a new dataset, one only needs to update the load() function in src/interpretability/data_loader.py to read the features (X_raw) for the new dataset from the dataset file. X_raw needs to have shape [N_d, D_f], where N_d is the number of examples in the dataset and D_f is the number of features for each example. Moreover, the code can also be easily used with additional CATE learners. We currently use the PyTorch implementations of the different CATE learners from the CATENets Python package: https://github.com/AliciaCurth/CATENets. To evaluate the ability of new/other CATE learners to discover predictive features for CATE estimation, one can add it to the learners' list in src/interpretability/experiments.py. The CATE learner needs to be implemented as a Python Class that inherits the BaseCATEEstimator in the CATENets package: https://github.com/AliciaCurth/CATENets/blob/main/catenets/models/torch/base.py#L514.
>
> ### What is the best practice to choose between CATE methods when attribution metrics and PEHE disagree?
>
> Thank you for this interesting question! We would argue that there is an inherent tradeoff here, whose resolution depends on the priorities/use-case the practitioner has in mind. If the end goal is simply to use a CATE estimator to predict treatment benefit for a patient, e.g. to decide whether or not to allocate treatment, a method with lower PEHE may be preferred. If the end goal is to identify a biomarker along which treatment benefit differs, e.g. to refine inclusion criteria for a clinical trial or prescription, a method with higher $Attr_{pred}$ may be preferred.

---

### Official Review · Reviewer_zRWR · 2022-07-22
**Generating ground truth for conditional average treatment effect (CATE)**

**Rating:** 6
**Confidence:** 3

**Strengths:**

The idea is extremely interesting and quite simple in itself. I like the quality of the explanations and motivation given by the authors, who underline the importance of analysing separately predictive and prognostic covariates. Indeed, benchmarks as such can only help pushing forward the research on causation.

**Weaknesses:**

The main weakness that I see in the paper is on the statement of the main contribution. It is unclear to me whether the authors are proposing a data-generation algorithm (i.e. Algorithm 1 on page 5), or if they are providing a benchmark of CATE learners that could be then extended to new learners and methods in the future. While the two things go hand in hand, I fear that the discussion on Algorithm 1 is not solid enough and that too much attention is given afterwards to the benchmarking results.


**Additional Feedback:**

I would like to stress again the need for a stronger discussion on the implication of generating arbitrary associations. I feel that the authors tried to avoid this discussion point that is instead at the very core of their contribution.
The authors could also introduce a small discussion on the limitations of Integrated Gradients (in particular with the baseline reference set to zeros).
Finally DGPs on page 5 was not defined before in the text.

**Clarity:**

The paper is clearly written and I could follow the structure and understand how to recreate their dataset.

**Correctness:**

I see the idea in principle as correct and I think it is quite simple and straightforward to implement. However, I do think that more attention should have been placed to the applicability of this method in high-risk domains such as the medical and biological domain. The authors do provide an example of a  bio-medical dataset with TCGA, but no discussion at all is given on the implications of their  arbitrary treatment assignments and outcomes in this context. TCGA contains several information on patient diagnostic, prognostic, and treatment. What aspect did the authors consider in the specific for this case? How do they think that their method could apply to similar datasets and what are the risks of using arbitrary associations to evaluate the CATE learners? This was my main concern during the entire read of the paper. What if the assigned outcome makes is really nonsense for the treatment that has been chosen? I think these kind of discussions must be in the paper.

**Documentation:**

Since the data generation is entirely synthetic, aspects on data collection, organization, availability and maintenance, ethical and responsible use are not a concern.


**Ethics:**

No ethical concerns

**Relation To Prior Work:**

Yes, the authors discuss the previous work and they clarify that their method, differently from the existing ones, sits on top of already trained models, proposing a post-hoc approach.

**Summary And Contributions:**

The authors propose a framework to generate data that provides us with ground-truth access to predictive covariates. The main claim of the paper is to use the proposed framework as a benchmark to evaluate CATE estimators. The authors underline the importance of distinguishing between predictive and prognostic covariates and they delineate a method to assess the identification quality of predictive covariates by multiple techniques.
They then show the results obtained on multiple dataset types (TCGA, Twins, News, ACIC2016) comparing the performance of five different methods.

---

> ### Author Response · Authors · 2022-08-14
> **References in the Response**
>
> ### References in the response
>
> [R1] Hill, Jennifer L. "Bayesian nonparametric modelling for causal inference." Journal of Computational and Graphical Statistics 20.1 (2011): 217-240.
>
> [R2] Vincent Dorie, Jennifer Hill, Uri Shalit, Marc Scott, and Dan Cervone. Automated versus do-it-yourself methods for causal inference: Lessons learned from a data analysis competition. Statistical Science, 34(1):43–68, 2019
>
> [R3] Hermansson, E., & Svensson, D. (2021). On Discovering Treatment-Effect Modifiers Using Virtual Twins and Causal Forest ML in the Presence of Prognostic Biomarkers. In International Conference on Computational Science and Its Applications (pp. 624-640). Springer, Cham.
>
> [R4] Sturmfels, P., Lundberg, S., & Lee, S. I. (2020). Visualizing the impact of feature attribution baselines. Distill, 5(1), e22.

---

> ### Author Response · Authors · 2022-08-14
> **Response to Reviewer zRWR**
>
> Thank you for your thoughtful comments and suggestions. We are grateful for your appreciation of our paper! We respond to your major points in turn below.
>
> ### Main contribution of the paper
>
> We thank the reviewer for allowing us to clarify this point! The main contribution of the paper is to provide and study a new (and important) interpretability axis to benchmark CATE learners on. This is the reason why the paper allocates much attention to the benchmarking results. In that light, the DGP proposed in Algorithm 1 is just a means to that benchmarking end. We emphasize that this DGP is built to be flexible enough to allow the user to simulate different difficulties that occur in real-world CATE estimation settings. This flexibility is implemented by letting the user specify the functional form of the prognostic prognostic part $\mu\_{\mathrm{prog}}$ and the predictive parts $\mu\_{\mathrm{pred}0}, \mu\_{\mathrm{pred}1}$ of our simulated potential outcomes. This allows us to benchmark the CATE models in 3 salient settings:
> - Experiment 1 benchmarks CATE models in a setting where treatment effects can be really small.
> - Experiment 2 benchmarks CATE models in a setting where treatment effects can be nonlinear functions of the covariates.
>  - Experiment 3 benchmarks CATE models in a setting where treatment assignment is not randomized.
>
> One important takeaway from our paper is that interpretability permits to compare CATE models beyond the standard PEHE analysis. We believe that the CATE estimation community would greatly benefit from incorporating this type of benchmark when introducing new CATE models.
>
> ### On the risk of using semi-simulated DGPs
>
> Allow us to emphasize that, due to the fact that counterfactuals are never observed, it is unfortunately not possible to obtain a meaningful evaluation of CATE estimation methods on real datasets. As such, throughout the causal inference literature on estimating potential outcomes, several different semi-synthetic data simulations have been widely used for evaluating PEHE, most prominently IHDP [R1] and ACIC2016 [R2]. Our benchmarking environment is partially inspired by these simulated benchmarks in that we randomly sample important features and outcome specifications (as they do) to determine responses; and then average our results over multiple realizations to ensure performance is evaluated _for many different_ feature-outcome relationships. (As different feature-outcome relationships may be more realistic for different applications, this should ensure that our results are relevant agnostic of application.)
>
> Nevertheless, while the DGP is synthetic, we would like to emphasize that it enables the simulation of datasets _with characteristics_ that can arise in realistic scenarios. In particular, we have shown that we can simulate settings where features have small predictive effects as in  [R3] by controlling the predictive scale, we simulate a wide range of non-linear outcome functions and different types of treatment selection bias (predictive confounding, prognostic confounding and non-confounded propensity), where the propensity scale can also be controlled in each setting.
>
> Because it will depend on the application at hand what kind of simulation scenario is most realistic, our investigation allowed us to study in more generality how performance differs across a range of possible scenarios, which can in turn guide a practitioner’s model choice when specific characteristics of the applications are known. Note also that, because our benchmark environment is modular, anyone with a specific application in mind could easily replicate our analysis using a different DGP, if that was more applicable to their setting of interest.
>
> ### Limitations of Integrated Gradients
>
> We see mainly two limitations for Integrated Gradients, both of which can be addressed in our setting.
> - To compute Integrated Gradients, the black-box needs to be differentiable with respect to its input features. In our setting this is not a problem since all learners are parametrized with feed–forward neural networks.
> - Integrated Gradients is sensitive to the choice of the baseline $\bar{x}$, as discussed in [R4]. For this reason, it is important to choose a baseline with a clear interpretation. In our setting, the baseline $\bar{x} = 0$ has a natural interpretation since all the covariates we manipulate are either normalized in the interval $[0, 1]$ (this is the case for the TCGA, News and Twins datasets) or standardized (this is the case for ACIC2016). Hence, the baseline corresponds to the minimal features in the former case and to the mean features in the later case.
>
> We have added the above discussion in the revised version of Appendix C.3.

---

> > ### Comment · Reviewer_zRWR · 2022-08-26
> > **Thanks for the clarifying comments**
> >
> > Thank you for clarifying my main concerns and for adding the discussion to the appendix!

---

### Official Review · Reviewer_HBNf · 2022-07-23
**Post-hoc feature importance to identify covariates that are important on CATE estimation.**

**Rating:** 6
**Confidence:** 4

**Strengths:**

1. It addresses an important causal inference problem: the evaluation of treatment effect methods.
2. The empirical study takes into consideration non-linearities and confounding.
3. Interesting distinction between predictive and prognostic features. Several applications could benefit from such distinction between covariates.
4. It allows the data simulation on top of existing datasets, which is a good and flexible characteristic.



**Weaknesses:**

1. The naming needs some improvement. It often mentions the 'discovery of covariates' which, considering the causality context, can be misinterpreted as a causal discovery problem. Important to point out/state the distinctions.  It also mentions confounding, but it is not clear how that would affect the identification (from the back-door perspective) of the treatment effects and feature importance. Are the confounders always observed?

2. The empirical study adopts a data generation process. While there are several plots to investigate if the methods work in these simulations, there is no analysis of how realistic they actually are on the real-world.

3. Considering the problem of CATE evaluation: ground truth is not available in real-world applications. Are there any guarantees of how confident can I be about the 'most important prognostic feature' being indeed the most important prognostic feature? What settings can we know for sure, and in which settings might fail?


**Additional Feedback:**

1) I suggest renaming the predictive covariates for something else.  Readers can easily misunderstand the term, as it is often used in the context of predictions (predictive covariable = use to predict Y) versus predictive of effect heterogeneity (as used on the paper).

2) The use of 'discovery of predictive covariates' is also not ideal. Causal Discovery is a distinct and harder problem, and a similar name could also mislead readers.

3)  5.1 Setup mentions alpha ~ Uniform(), but that is not shown in algorithm 1.

4) Maybe standardize the Attr_pred and Att_prod. From my understanding, one is good if close to 1, and the other, if close to 0. It is easier to understand and interpret results if the ideal solution is on the same scale (close to 1 is good for both).

5) You mentioned that PEHE is not directly comparable (lines 266-267). So maybe use a different plot, as that is not clear with the current line plots.

6) The conclusion of 5.2 is not clearly stated. Does that mean the proposed approach fails if there are nonlinearities or confounding? If yes, that should be more clearly stated as a model limitation.


For reference: I spent around 6h reading the paper, code, and appendix.


POST-REBUTTAL
Several of my concerns were addressed or explained by the authors. I increased my score.
I still think the writing could be improved, but most of my concerns regarding the method itself were clarified.


**Clarity:**

The paper is very hard to read. I had to re-read several paragraphs to understand what the author was describing.

Some terms are now well defined. For example, what "discover drivers" means? Are these similar to causal discovery?

A formal definition of prognostic and predictive covariates could also improve the paper.

Figure 1 is very hard to read, the letters are very small. I have printed the paper, and I can not read anything. Other plots also have very small letters (but are more legible).

The author use circle numbers several times, and it is confusing. Maybe keep to the most important numbered items, and find a better way to enumerate the other items with words.

The discussion of why CATE is special is not clear. It seems the author is just describing how it is setting up the experiments again.

**Correctness:**

The paper contains an empirical study to evaluate a data generation process on top of existing datasets. The goal is to estimate CATE.

In the paper checklist, the authors claim that they didn't state all assumptions and theoretical results because their papers are purely empirical. However, I think some assumptions are required to offer some guarantees. Based on the experiments, the proposed method performs poorly if there are nonlinearities or confounding.  What does that mean in practice? If I have an application for such a method, and there are nonlinearities and confounding, can I apply the method and trust 100% the feature importance that it outputs? My intuition says that, while the empirical study worked on simple examples, that might not be true for other more complex applications, and results should take interpreted very carefully. Hence, assumptions regarding the covariates or a clear description of their limitations would be very welcoming.

Hence, while not necessarily incorrect, I would say the paper is incomplete and lacks a proper discussion on limitations, assumptions, and interpretation of empirical results.

**Documentation:**

The code to run the experiments is available and seems well written. It contains requirements and seeds for reproducibility.

The readme.md could be more descriptive. It contains the code to re-run the paper results. However, there is no mention of how I would use the method on a new dataset or with different methods. Is the current implementation suitable only to re-run the experiments on the paper?

I would expect that a paper that proposes an approach to make simulations on top of existing datasets would offer an easy way to add new datasets to the pipeline and adopt different CATE estimators. However, that doesn't seem to be the case.


**Ethics:**

Not applicable.

**Relation To Prior Work:**

The author mentions some related works on feature importance, and approaches to interpreting CATE.

Maybe the authors could have adopted some datasets that are more complex. What happens with the datasets being high-dimensional? Or if the covariates are images? If the method does work with these other datasets, it is one more indication that maybe some assumptions or a complete description of the limitations are missing.

**Summary And Contributions:**

This paper proposes a DGP that allows post-hoc feature importance, used to identify covariates important on CATE estimation.

The feature important is especially useful if the CATE estimator is a 'black-box' (Neural network).

The feature importance method was Integrated Gradients due to performance and efficiency (other methods available in Appendix).

The proposed DGP was evaluated empirically on several datasets, with several CATE estimators.

---

> ### Author Response · Authors · 2022-08-14
> **Response to Reviewer HBNf (Part 5/5)**
>
> ### Additional Feedback
> - _No mention of alpha in Algorithm 1._ In Algorithm 1, we did not specify a fixed functional form for the prognostic prognostic part $\mu\_{\mathrm{prog}}$ and the predictive parts $\mu\_{\mathrm{pred}0}, \mu\_{\mathrm{pred}1}$ of our simulated potential outcomes. This is because our semi-synthetic data generating process is flexible and these functions can be adapted to study specific properties. For instance, the parametrization from Experiment 1 permits to adjust the treatment effect strength through the additional parameter $\omega_{\mathrm{pred}}$.  We note that Algorithm 1 is consistent with the above discussion as the potential outcome functions (which we parametrize using alpha as a special case) are given as *inputs* in the Algorithm.
> - _Choice for the metrics $\mathrm{Attr}\_{\mathrm{pred}}$ and $\mathrm{Attr}\_{\mathrm{prog}}$_ We confirm that the reviewer's understanding is correct. We use $\mathrm{Attr}\_{\mathrm{pred}}$ and $\mathrm{Attr}\_{\mathrm{prog}}$ because they both correspond to quantities that are easy to interpret: the fraction of the CATE model’s prediction attributed to the predictive and prognostic covariates, respectively. To retain the meaning of these quantities, we believe that it is preferable to keep their current form, even if one metric should be maximized and the other should be minimized. In order to clarify this point in the revised manuscript, we have annotated Figures 2 and 3 to indicate whether higher/lower values are better, as is often done in the literature when metrics have opposite interpretations (e.g. AUC/accuracy vs MSE/MAE).
> - _Comparability of PEHE across scenarios._ Please note that the reason why we mention that the PEHE is not directly comparable across different values of $\omega_{\mathrm{pred}}$ is that the magnitude of the simulated outcomes (and thus the CATE) also increases as we increase $\omega_{\mathrm{pred}}$. Because of this, a larger PEHE for higher $\omega_{\mathrm{pred}}$ is expected and should not be used to compare the performance of a single learner across $\omega_{\mathrm{pred}}$ (which we do not do). However, because we are interested in comparing different learners at the same value of $\omega_{\mathrm{pred}}$, using PEHE is not a problem. Note also that PEHE is usually indeed used in the literature without regard for changes in CATE magnitude
> - _"The conclusion of 5.2 is not clearly stated. Does that mean the proposed approach fails if there are nonlinearities or confounding? If yes, that should be more clearly stated as a model limitation."_  Allow us to emphasize once more that we propose a new dimension of evaluation for CATE learners (their ability to discover predictive covariates) precisely with the goal of _highlight limitations_ of existing learning strategies.  In our experiments, we find that all existing methods deteriorate in the presence of stronger nonlinearities and confounding, thus their discoveries should be considered with caution in practice.

---

> > ### Comment · Reviewer_HBNf · 2022-08-16
> > **Thanks for the extensive comments**
> >
> > Thanks for addressing/commenting on all my concerns. It helped to clarify some of the terms and better understand the paper.
> >
> > I raised my score to reflect my new opinion on the paper. I still think the writing could be improved in some parts, mostly to improve the flow/understanding of the paper. However, the method seems ok, consistent with the literature, the experiments are convincing, code is good.

---

> ### Author Response · Authors · 2022-08-14
> **Response to Reviewer HBNf (Part 4/5)**
>
> ### Using more complex datasets [ct'd]
>
> Therefore, we have now performed additional experiments where we have increased the scale of our datasets in terms of the dimensionality of the feature space. In particular, we have obtained a version of the TCGA data which has the $5000$ most variable genes as features (instead of 100 as used in our initial experiments). Out of the $d=5000$ features, we set $n_{\mathcal{I}} = 100$ of them to be important as prognostic and predictive for each potential outcome. We report here:  https://imgur.com/a/92AErOd the performance comparison in terms of $Attr_{pred}$ (left, higher is better), $Attr_{prog}$ (middle, lower is better) and PEHE (right, lower is better) when varying the predictive scale. We have also included a new section in the paper to discuss the results in Appendix C.4. We find that in this setting, the different learners have poor absolute performance in discovering the predictive features, while relative performance remains similar to the smaller scale experiments. Thus, our benchmark suggests that developing CATE learners that can better identify predictive covariates in a high dimensional setting is needed and would represent an interesting direction for future work.
>
> Note also that the vast majority of the CATE estimation literature is currently focussing on tabular data as input data. Considering e.g. images as input would therefore completely change the problem under consideration away from what is expected in the literature; this is therefore not a limitation of our paper but rather the CATE estimation literature as a whole.
>
> [R10] Schwab, Patrick, et al. "Learning counterfactual representations for estimating individual dose-response curves." Proceedings of the AAAI Conference on Artificial Intelligence. Vol. 34. No. 04. 2020.
>
> [R11] Kaddour, Jean, et al. "Causal effect inference for structured treatments." Advances in Neural Information Processing Systems 34 (2021): 24841-24854.
>
> [R12] Yoon, Jinsung, et al. "GANITE: Estimation of individualized treatment effects using generative adversarial nets." International Conference on Learning Representations. 2018.
>
> [R13] Zhang, Yao, et al. "Learning overlapping representations for the estimation of individualized treatment effects." International Conference on Artificial Intelligence and Statistics. PMLR, 2020.
>
> [R14] Johansson, Fredrik, Uri Shalit, and David Sontag. "Learning representations for counterfactual inference." International conference on machine learning. PMLR, 2016.
>
> [R15] Curth, Alicia, and Mihaela van der Schaar. "On inductive biases for heterogeneous treatment effect estimation." Advances in Neural Information Processing Systems 34 (2021): 15883-15894.
>
> ### Code documentation: how to use new datasets or learners?
> Thank you very much for your comment! Our code has indeed been written such that it could be easily extended by adding a new dataset or a new learner for evaluation. We have now updated the documentation in README.md to specifically mention how to do so.
>
> In particular, to add a new dataset, one only needs to update the load() function in src/interpretability/data_loader.py to read the features (X_raw) for the new dataset from the
> dataset file. X_raw needs to have shape [N_d, D_f], where N_d is the number of examples in the dataset and D_f is the number of features for each example. Moreover, the code can also be easily used with additional CATE learners. We currently use the PyTorch implementations of the different CATE learners from the
> CATENets Python package: https://github.com/AliciaCurth/CATENets. To evaluate the ability of new/other CATE learners to discover predictive features for CATE estimation, one can add it to the learners list in src/interpretability/experiments.py. The CATE learner needs to be implemented as a Python Class that inherits the BaseCATEEstimator in the CATENets package: https://github.com/AliciaCurth/CATENets/blob/main/catenets/models/torch/base.py#L514.

---

> ### Author Response · Authors · 2022-08-14
> **Response to Reviewer HBNf (Part 3/5)**
>
> ### Formal definition of prognostic and predictive covariates (Clarity)
>
> Prognostic and predictive covariates are defined in Section 3 according to the relevant literature [R8,R9]. If we assume that the potential outcomes $\mu\_0 , \mu\_1$ are differentiable with respect to the  covariates, one can provide a more formal definition. We say that a feature $x_i$ is *prognostic* if both potential outcomes depend on that feature: $\frac{\partial \mu_0}{\partial x_i} \neq 0$ and $\frac{\partial \mu_1}{\partial x_i} \neq 0$. We say that a feature $x_i$ is *predictive* if the CATE depends on that feature: $\frac{\partial \tau}{\partial x_i} \neq 0$. One can trivially check that the prognostic and predictive features from Experiments 1, 2 and 3 are indeed consistent with the above definitions. We have added this more formal definition in Section 3 of the revised manuscript.
>
> [R8] Karla V Ballman. Biomarker: predictive or prognostic? Journal of clinical oncology: official
> journal of the American Society of Clinical Oncology, 33(33):3968–3971, 2015.
>
> [R9] Konstantinos Sechidis, Konstantinos Papangelou, Paul D Metcalfe, David Svensson, James Weatherall, and Gavin Brown. Distinguishing prognostic and predictive biomarkers: an information theoretic approach. Bioinformatics, 34(19):3365–3376, 2018.
>
> _Renaming predictive covariates?(Additional Feedback)_ Note that the term ‘predictive covariate’ is well-established in the literature [R8, R9], and specifically popular in the medical literature on this topic. We will therefore continue to use it in our work.
>
> ### Other comments on clarity
> _Figure 1 size._ We thank the reviewer for pointing this out. The resolution of Figure 1 has been improved in the revised manuscript.
>
> _Use of circled numbers._ We thank the reviewer for highlighting that some readers may find this stylistic device confusing if used too much. We have updated the manuscript and now only use the circled numbers in Section 2 to label the different CATE characteristics, and in Section 5 to highlight the different experimental findings.
>
> _"The discussion of why CATE is special is not clear. It seems the author is just describing how it is setting up the experiments again"_. The purpose of the Section “What makes CATE Estimation Special?” was to highlight to readers unfamiliar with the treatment effect context how CATE estimation differs from _standard supervised learning_. To make this more clear, we have updated the title of the Section to read  “What makes CATE Estimation Special relative to Supervised Learning?”
>
> Note that we indeed purposefully set up the section on “What makes CATE Estimation Special?” to parallel the experimental section to *motivate* why the CATE estimation context led us to set up our experiments the way that we did. We think that this could be particularly helpful for readers unfamiliar with the CATE estimation context as they may only be used to experimental evaluations in the standard supervised learning context, which usually look quite different due to the differences in problem setup.
>
> ### Using more complex datasets
> Please note that in our experiments we use datasets that are standard throughout the causal inference literature for CATE estimation that range in size from 2200 (ACIC2016) to 11400 examples (Twins). The features from the datasets we used in our benchmark have been commonly used in semi-synthetic set-ups for evaluation by several CATE estimation methods: TCGA [R10, R11], Twins [R12, R13], News [R10, R14], ACIC2016 [R15]. In fact, the Twins dataset is one of the larger-scale datasets (in terms of the number of data points) commonly used throughout the literature.
>
> Nevertheless, we would like to emphasize once more that in this paper we are proposing a new _dimension_ for evaluating CATE models in terms of their ability to discover the correct predictive covariates when estimating treament effects. The benchmark environment that we have developed for evaluating this new dimension can be easily extended to incorporate additional datasets and models. In particular, our proposed Data Generating Process (DGP) can be used with any covariates dataset as input (see Algorithm 1), thus also with higher dimensional datasets.
>
> [Continued in next comment]

---

> ### Author Response · Authors · 2022-08-14
> **Response to Reviewer HBNf (Part 2/5)**
>
> ### Realism of the data generation process (Weakness 2) [ct'd]
> Because it will depend on the application at hand what kind of simulation scenario is most realistic, our investigation allowed to study in more generality how performance differs across a range of possible scenarios, which can in turn guide a practitioner’s model choice when specific characteristics of the applications are known. Note also that, because our benchmark environment is modular, anyone with a specific application in mind could easily replicate our analysis using a different DGP, if that was more applicable to their setting of interest.
>
> [R4] Hill, Jennifer L. "Bayesian nonparametric modelling for causal inference." Journal of Computational and Graphical Statistics 20.1 (2011): 217-240.
>
> [R5] Vincent Dorie, Jennifer Hill, Uri Shalit, Marc Scott, and Dan Cervone. Automated versus do-it-yourself methods for causal inference: Lessons learned from a data analysis competition. Statistical Science, 34(1):43–68, 2019
>
> [R6] Ismail, A. A., Gunady, M., Corrada Bravo, H., & Feizi, S. (2020). Benchmarking deep learning interpretability in time series predictions. Advances in neural information processing systems, 33, 6441-6452.
>
> [R7] Yoon, J., Jordon, J., & van der Schaar, M. (2018). INVASE: Instance-wise variable selection using neural networks. In International Conference on Learning Representations.
>
> ### Extending the insights from our paper to real-world applications (Weakness 3)
>
> The absence of ground-truth in real-world application is a general difficulty in the CATE estimation setting and the reason why many papers in the litterature, including ours, incorporate a synthetic or semi-synthetic setting. Although our potential outcomes $\mu\_0$ and $\mu\_1$ are synthetically generated, their functional forms are motivated by 3 challenges that occur in real-world CATE estimation:
>
> - Experiment 1 is motivated by the fact that treatment effects can be really small in practice.
> - Experiment 2 is motivated by the fact that treatment effects can be nonlinear functions of the covariates.
> - Experiment 3 is motivated by the fact that treatment assignment is not necessarily randomized in observational data.
>
> Each of these experiments consistently demonstrate that these real-world difficulties negatively impact the ability of _all_ CATE estimators under investigation to identify predictive covariates. In that sense, our paper suggests that the user should not always trust the predictive covariates surfaced by a CATE model in a setting with weak/highly nonlinear treatment effect or confounding. Note that our contribution is *not* to propose a new CATE estimation method that addresses all of those real-world challenges but rather to use interpretability to study the effect of those real-world challenges on existing CATE estimators. One important takeaway from our paper is that interpretability permits to compare CATE estimators beyond the standard PEHE analysis. We believe that the CATE estimation community would greatly benefit from incorporating the proposed type of analysis when introducing new CATE models.
>
> ### Paper checklist: assumptions required to offer guarantees (Correctness)
>
> Please note that the referenced part of the paper checklist (Q2a/b) is explicitly required only “If you are including theoretical results. “. This is not the case for our paper, hence our completion of the checklist is correct.
>
> We do agree that assumptions may be required to give theoretical guarantees, however, deriving such guarantees would go beyond the scope of the current paper, which empirically investigates a new dimension of evaluation of CATE estimators. (Note again that the goal of this paper was not _to propose a new method_, but to empirically investigate how combinations of existing CATE estimators and interpretability methods fare at the new task of discovering predictive covariates. In our experiments we indeed already highlighted possible failure modes of these existing methods.)
>
> To make this even more clear, we have updated the discussion section to invite future work studying failure modes of methods, i.e. to both (i) consider an even wider range of possible DGPs to investigate how our results generalize empirically and (ii) study the problem of predictive covariate discovery theoretically.

---

> ### Author Response · Authors · 2022-08-14
> **Response to Reviewer HBNf (Part 1/5)**
>
> Thank you for your thoughtful comments and suggestions. We are happy to hear that you believe several applications could benefit from the concepts studied in our work. Below, we respond to your major comments in turn. We have attempted to follow the order in the review, but have grouped responses to related comments where applicable.
>
> ---------------------------------------
> ### Naming: 'Discovery of covariates' vs 'causal discovery' _(Weakness 1)_
>
> We would like to emphasize that our usage of the term “(predictive) covariate discovery” is consistent with the relevant literature on feature selection and feature importance. See for instance [R1] and [R2] that discuss the usage of feature selection/importance for *biomarker discovery*. In the same way, we use feature importance to assess the ability of CATE models to *discover predictive covariates*, in line with [R3], in a setting where the ground-truth is known. Hence, our notion of *discovery* is different from the one used in the literature on *causal discovery*, since we are not interested in discovering the causal graph underlying the data generating process. As we make no mention of the literature on graph-based causality, we do not think that the notion of discovery can be misinterpreted in our paper. To the contrary, we believe that explicit mention of causal discovery in our paper, even for the purpose of disambiguation, may do more harm than good and confuse readers into thinking that our paper is more related to causal discovery than it is.
>
> [R1] He, Z., & Yu, W. (2010). Stable feature selection for biomarker discovery. Computational biology and chemistry, 34(4), 215-225.
>
> [R2] Huynh-Thu, V. A., Saeys, Y., Wehenkel, L., & Geurts, P. (2012). Statistical interpretation of machine learning-based feature importance scores for biomarker discovery. Bioinformatics, 28(13), 1766-1774.
>
> [R3] Hermansson, E., & Svensson, D. (2021). On Discovering Treatment-Effect Modifiers Using Virtual Twins and Causal Forest ML in the Presence of Prognostic Biomarkers. In International Conference on Computational Science and Its Applications (pp. 624-640). Springer, Cham.
>
> ### The presence of confounders (Weakness 1)
>
> We make the assumption that there are no unobserved confounders, as stated in Assumption 1 in  Appendix A. The statement of the assumption is standard within the treatment effect estimation literature in the potential outcomes framework, and corresponds to the assumption that the covariates X jointly block all backdoor paths into Y.  We have added an explicit reference to the appendix containing these assumptions in Section 2 of the revised manuscript.
>
> ### Realism of the data generation process (Weakness 2)
>
> Please note that due to the fact that counterfactuals are never observed, it is not possible to obtain a meaningful evaluation of CATE estimation methods on real datasets. As such, throughout the causal inference literature on estimating potential outcomes, several different semi-synthetic data simulations have been widely used for evaluating PEHE, such as IHDP [R4] and ACIC2016 [R5].
>
> In our paper, we propose a new dimension of evaluation for these methods in terms of their ability to discover the correct predictive covariates when estimating the treatment effects. This creates an additional challenge for evaluation, as we now also need ground-truth knowledge of these predictive covariates [R6, R7] which would usually not be available in real-datasets.
>
> Consequently, access to a DGP with simulated outcomes and known predictive and prognostic covariates is **needed** for a meaningful evaluation in our setting. Nevertheless, while the DGP is synthetic, we would like to emphasize that it enables the simulation of datasets _with characteristics_ that can arise in realistic scenarios. In particular, we have shown that we can simulate settings where features have small predictive effects as in  [R3] by controlling the predictive scale, we simulate a wide range of non-linear outcome functions and different types of treatment selection bias (predictive confounding, prognostic confounding and non-confounded propensity), where the propensity scale can also be controlled in each setting.
>
> [Continued in comment below]

---

### Official Review · Reviewer_rxGH · 2022-07-24
**Well-written and designed benchmark paper. Motivation and empirical evaluation could be improved.**

**Rating:** 8
**Confidence:** 3
**Correctness:** Yes, everything looks designed and pe…

**Strengths:**

* The paper is well-written, and the benchmark design looks sensible and is easy to follow
* The experiments reveal novel insights compared to an evaluation based solely on PEHE. Therefore, this contribution can become significant and relevant to the broader research community.


**Weaknesses:**

# Pre-Rebuttal
A common CATE meta-learning strategy is to incorporate a nuisance estimator for the prognostic effect $m(x) = E[Y | X=x]$. For example, see the works of [1] using non-neural-network models or [2] using NNs. The idea behind this strategy is to construct a pseudo-outcome in which the prognostic effect was removed, such that the CATE estimator can better identify the predictive covariates.

I think adding at least one method that constructs a pseudo-outcome with removed (estimated) prognostic effect to the evaluation would be appropriate. Unfortunately, as far as I can tell, the authors have not evaluated such an estimator/strategy, which is precisely motivated for settings where there can be non-predictive covariates that we want to regularize against.

Another minor weakness is that I think the motivation for such settings with naturally prevalent non-predictive covariates in the dataset could be better explained. It is not clear to me in which settings many non-predictive covariates naturally occur that a data scientist would not be able to filter out a priori. The above-mentioned works may be partly used as motivation, but I think introducing a novel benchmark and convincing future researchers to invest their time to use it requires a strong, watertight motivation for such settings. One way to explain this better can be to point out two or three concrete examples with many non-predictive covariates motivated through real-life scenarios.

I don't find the cited drug discovery example mentioned in the introduction very clear. While it says that it is important that a model discovers the correct drivers of the underlying effect heterogeneity, it is not apparent to me why this is particularly important in drug discovery or why in these datasets, there may be many non-predictive covariates.


[1] Nie et al, Quasi-Oracle Estimation of Heterogeneous Treatment Effects
[2] Kaddour et al, Causal Effect Inference for Structured Treatments

# Post-rebuttal
The above concerns have been addressed and I raised my score.

**Additional Feedback:**

No additional feedback

**Clarity:**

Yes, overall, the paper is indeed very well-written and easy to follow, besides the motivation for the problem

**Documentation:**

Yes, I had a quick look at the attached code and it looks good to me.

**Ethics:**

No, the authors re-use covariates from existing dataset and contribute synthetical outcome functions which do not raise any ethical concerns

**Relation To Prior Work:**

The paper discusses previous benchmarks and issues thereof but does not mention works constructing a pseudo outcome with removed prognostic effect, see


[1] Nie et al, Quasi-Oracle Estimation of Heterogeneous Treatment Effects
[2] Kaddour et al, Causal Effect Inference for Structured Treatments


**Summary And Contributions:**

The estimation of conditional average treatment effect (CATEs) is the problem of predicting $\tau(\mathbf X)$, the difference between the expected outcomes of an individual/sub-group, characterized by covariates $\mathbf X$ under different treatments. A common assumption in the literature is that not all covariate features are relevant for the CATE. The authors propose to evaluate the ability of CATE estimators to distinguish between predictive and non-predictive covariates. For that, they propose semi-synthetic simulations based on existing covariate features. In their evaluations, they compare the ability of existing NN-based CATE estimators to identify the predictive covariates under varying strengths of predictive effects, nonlinearities, and confounding.

---

> ### Author Response · Authors · 2022-08-14
> **References in the response**
>
> ### References in the response
> [R1] Künzel, S. R., Sekhon, J. S., Bickel, P. J., & Yu, B. (2019). Metalearners for estimating heterogeneous treatment effects using machine learning. Proceedings of the national academy of sciences, 116(10), 4156-4165.
>
> [R2] Kennedy, E. H. (2020). Optimal doubly robust estimation of heterogeneous causal effects. arXiv preprint arXiv:2004.14497.
>
> [R3] Nie, X., & Wager, S. (2021). Quasi-oracle estimation of heterogeneous treatment effects. Biometrika, 108(2), 299-319.
>
> [R4] Cinelli, C., Forney, A., & Pearl, J. (2021). A crash course in good and bad controls. Sociological Methods & Research, 00491241221099552.
>
> [R5] Karla V Ballman. Biomarker: predictive or prognostic? Journal of clinical oncology: official journal of the American Society of Clinical Oncology, 33(33):3968–3971, 2015.
>
> [R6] Hingorani, A. D., van der Windt, D. A., Riley, R. D., Abrams, K., Moons, K. G., Steyerberg, E. W., ... & Hemingway, H. (2013). Prognosis research strategy (PROGRESS) 4: stratified medicine research. Bmj, 346.
>
> [R7] Clark, G. M. (2008). Prognostic factors versus predictive factors: examples from a clinical trial of erlotinib. Molecular oncology, 1(4), 406-412.
>
> [R8] Hermansson, E., & Svensson, D. (2021, September). On Discovering Treatment-Effect Modifiers Using Virtual Twins and Causal Forest ML in the Presence of Prognostic Biomarkers. In International Conference on Computational Science and Its Applications (pp. 624-640). Springer, Cham.
>
> [R9] Dmitrienko, A., Lipkovich, I., Dane, A., & Muysers, C. (2020). Data-driven and confirmatory subgroup analysis in clinical trials. In Design and Analysis of Subgroups with Biopharmaceutical Applications (pp. 33-91). Springer, Cham.

---

> ### Author Response · Authors · 2022-08-14
> **Response to Reviewer rxGH**
>
> Thank you for your thoughtful comments and suggestions. We are grateful for your appreciation of our paper! We respond to your major points in turn below.
>
> ### Removing prognostic effects using the R-learner
>
> First, allow us to emphasize that both X- and DR-learner – which we already use in our experiments –  *do* operate in the manner described in the review: as we discuss in Appendix C1, both learners use a first-stage nuisance estimation step to construct a pseudo-outcome and then use this as a target in a second-stage direct regression for CATE – which indeed effectively aims to remove any prognostic effect (and is how these learners were originally motivated, see [R1, R2]). Note also that the first-stage outcome component of the R-learner,$\mu(x)=\mathbb{E}[Y|X=x]$, does not condition on treatment assignment and is therefore equal to $\mu(x) = \mu_0(x)+\pi(x)\tau(x)$ [R3] – i.e. it requires estimation of *more than just the prognostic component*.
>
> We have replicated our experiments of Section 5.1 using the R-learner, a plot of which can be found here: https://imgur.com/a/nfDc1Nm. We find that the R-learner indeed performs almost identically to X- and DR-learner in terms of correctly identifying predictive covariates, but sometimes performs slightly better in removing prognostic covariates when $\omega_{pred}$ is small. At the same time, we also find that it can sometimes perform slightly worse in terms of PEHE when $\omega_{pred}$ is large. Due to the similarity in motivation and performance between R-learner and DR- & X-learner, we would prefer to avoid clutter in our plots and not include this additional learner. We have however updated the description of learners in appendix C1 to include the relevant citations mentioned in the review (thank you!).
>
> ###  In which settings do many non-predictive covariates naturally occur that a data scientist would not be able to filter out a priori?
>
> First, allow us to argue that it is generally not desirable to filter out _all_ non-predictive covariates a priori: while non-predictive covariates which are irrelevant (ie also not prognostic) can be excluded if such knowledge is available,  non-predictive covariates which are prognostic _should_ be included in treatment effect analyses (i) if they are confounders as the estimates will otherwise be biased and (ii) even if they are not confounders, including them will generally lead to improved precision of estimators [R4]. Additionally, even if all predictive covariates were known a priori, most existing CATE learners studied in our paper would not allow to make use of that knowledge: only two stage estimation strategies such as X-, R- and DR-learner straightforwardly allow to restrict whether a subset of covariates enters the CATE estimator.
>
> Second, much less is generally known about predictive covariates than about (prognostic) risk factors for disease. One reason for this is that prognostic covariates can be established by considering any available sample of untreated individuals, while establishing predictive factors for a new treatment requires comparison between treatment and control group [R5], for which existing trial data may be underpowered – so evidence often appears gradually, from secondary and meta-analyses of existing trials over time [R6] . Prominent examples for predictive biomarkers include HER-2 status in breast cancer for the use of trastuzumab, the BCR-ABL mutation in chronic myeloid leukemia for the use of imatinib and EGFR mutations in pulmonary adenocarcinoma for the use of gefitinib [R6]. Further,  [R7] gives an example where predictive and prognostic covariates have been mixed up by experts: they show that in non-small cell lung cancer, treatment decisions for prescribing EGFR inhibitors have been made based on gender, histology and smoking history of patients, while gender and histology are actually prognostic covariates and hence are not actually predictive of treatment response.
>
>
> ### “Drug discovery” example in the introduction.
>
> Note that the example in the introduction is not about *drug discovery* but rather about *drug development* (as used in the pharmaceutical literature to describe also the process in which a drug passes through clinical trial phases). As detailed in [R8, R9] , discovery of effect modifying covariates is important both in exploratory trial phases – to identify potential biomarkers which may help to explain an enhanced treatment effect for future development – and  in confirmatory trial phases – to rule out that a drug is ineffective for some biomarker signatures or to otherwise adapt prescription criteria. We have expanded on the example in the introduction, including the examples above,  and hope that this has made it more clear!  We also note that it is considered a well-known challenge in such clinical trial datasets that there are many prognostic (i.e. non-predictive) covariates [R9], e.g. age, gender and other known risk factors for a given disease.

---

> > ### Comment · Reviewer_rxGH · 2022-08-16
> > **Thank you for your detailed response.**
> >
> > Thank you for your response.
> >
> > I was aware of the X- and DR-learner constructing a pseudo-outcome targeting the treatment effect. However, I misunderstood the purpose of the R-learner's conditional outcome model $m(x)$. Now, I realize that the R-learner is not necessarily better positioned to solve your proposed problem. Also, thank you for providing these additional plots that confirm this empirically.
> >
> > I raised my score.

---

### Official Review · Reviewer_yDEt · 2022-07-26
**Exciting work on interpretability of heterogeneous treatment effect models**

**Rating:** 7
**Confidence:** 3
**Correctness:** Yes

**Strengths:**

1. Estimation of post hoc feature importance in machine learning methods is an important and challenging task, especially in the healthcare domain.
2. Incorporation of CATE in machine learning models is a hot topic and the authors propose a novel benchmark that aims to identify predictive and prognostic covariates using CATE. This is important to push the field forward and therefore, this work is appreciated.
3. The motivation and the related works are explained clearly.
4. The authors have implemented a novel data generation method for their task which is very relevant for the proposed benchmark.
5. The authors have explored various learners on several datasets as well as several tasks that show the completeness of this benchmark.
6. The hyperparameters used are made public ensuring reproducibility of their analysis.

**Weaknesses:**

### PRE-REBUTTAL
1. It is not clear why the authors used an 80%-20% split to train and test CATE estimators. Cross-validation would be a more robust strategy to apply in this setting.
2. Figure 1 is not clear. A better resolution image would make it easier to read.
3. There are minor typing errors in the text - for example, on page 1 (Introduction), "an individuals potential outcomes" should be "an individual's potential outcomes".

### POST-REBUTTAL
The above issues have been resolved and appropriate justifications have been provided.

**Additional Feedback:**

-

**Clarity:**

Yes, although some concepts were difficult to understand based on the text and required external reading.

**Documentation:**

Yes

**Ethics:**

No ethical concerns.

**Relation To Prior Work:**

Yes

**Summary And Contributions:**

The authors propose a benchmark called ITErpretability that focuses on post hoc feature importance methods for black-box predictive models. More specifically, the authors explore the estimation of conditional average treatment effects (CAGE) to identify important features (covariates) that a given Machine Learning (ML) model uses. A novel data generation strategy is used by the authors to distinguish between predictive and prognostic covariates. Finally, various experiments are explored by the authors to test various CAGE estimators on the basis of their effect on strength of predictive effects, in the nonlinear setting, and on the effect of compounding on them.

---

> ### Author Response · Authors · 2022-08-14
> **Response to Reviewer yDEt**
>
> Thank you for your thoughtful comments and suggestions. We are grateful for your appreciation of our paper! We respond to your points in turn below.
>
> ### Train-test split vs. cross-validation
> Allow us to emphasize that using a train-test split for evaluation is standard across the causal inference literature for CATE estimation [R1, R2, R3, R4], which is why we also adopted this strategy in our experiments. Moreover, to obtain robust results, we ensure enough variability across the different seeds used for the experiments.
> We have used 30 different random seeds for each of the 4 datasets and for each setting of the predictive scale $\omega_{\mathrm{pred}} \in \{10^{-3}, 10^{-2}, 10^{-1}, 0.5, 1\}$ and non-linearity scale $\omega_{\mathrm{nl}} \in \{0.0, 0.2, 0.5, 0.7, 0.9, 1.0\}$  and 10 different random seeds for each of the 4 datasets, for each of the predictive confounding, prognostic confounding  and non-confounded propensity settings when changing the $\omega_\pi \in \{0, 0.5, 1, 2, 5, 10\} $. For each random seed, we sample different features from the dataset to be predictive and prognostic thus resulting in external variation through usage of a different data generating process (DGP) and different ground truth features for computing the evaluation metrics.
>
> [R1] Johansson, Fredrik, Uri Shalit, and David Sontag. "Learning representations for counterfactual inference." International conference on machine learning. PMLR, 2016.
>
> [R2] Shalit, Uri, et al. "Estimating individual treatment effect: generalization bounds and algorithms." International Conference on Machine Learning. PMLR, 2017.
>
> [R3] Yoon, Jinsung, et al. "GANITE: Estimation of individualized treatment effects using generative adversarial nets." International Conference on Learning Representations. 2018.
>
> [R4] Zhang, Yao, et al. "Learning overlapping representations for the estimation of individualized treatment effects." International Conference on Artificial Intelligence and Statistics. PMLR, 2020.
>
> ### Figure 1 resolution
> Thank you for pointing out issues with Figure 1. The resolution of Figure 1 has been improved in the revised manuscript!

---

### Official Review · Reviewer_MjU8 · 2022-07-28
**Review for Benchmarking Heterogeneous Treatment Effect Models through the Lens of Interpretability**

**Rating:** 7
**Confidence:** 3
**Clarity:** Paper is clearly written.

**Strengths:**

- The classification of features into predictive and prognostic in CATE estimation context are presented.
- Data generating process along with the metric for effect heterogeneity drivers identification are detailedly explained.
- Experimental results including variety of realistic setups are clearly identified.

**Weaknesses:**

- Simulations based on large-scale datasets are missed.
- Lack of hyperparameter tuning in experiments.

**Additional Feedback:**

- Adding large-scale real datasets could help to better distinguish between the CATE estimators in terms of predictive features identification and bring the new insights into performance of existing methods.
- Line 342: In a light of your results, how to explain the CFRNet’s significant outperformance of the baselines in the original paper (Shalit, 2017), on the IHDP and Jobs datasets ? In particular, does it mean that the treatment assignment bias has a different nature from ones simulated in your work ?

Small typos:
- Line 249: should it be “high Attr_{prog}” instead of “low Attr_{prog}” ?
- Line 251: “does does”

**Correctness:**

The evaluation methods and experiment design are appropriate and performed correctly.

**Documentation:**

There is sufficient detail to support reproducibility.

**Ethics:**

No.

**Relation To Prior Work:**

Related work in Section 1 clearly describes the difference of this work from previous contributions.

**Summary And Contributions:**

The paper contributes to the domain of causal inference, in particular to the area of Heterogeneous Treatment Effect (HTE) estimation. Authors propose a methodology to identify the drivers of treatment effect heterogeneity along with a corresponding semi-synthetic benchmark and the new insights into the performance of existing methods on the identification task.

---

> ### Author Response · Authors · 2022-08-14
> **Response to Reviewer MjU8: Part 2/2**
>
> ### Why does CFRNet perform differently here than in the original paper?
>
> Thank you for this interesting question! We think there are multiple rationales for why this may be the case:
> - First, note that CFRNet actually only outperforms TARNet on the IHDP dataset in the original paper(whereas on Jobs they perform equivalently.) As shown in [R9], the IHDP benchmark has a somewhat peculiar structure, where a small number of the 1000 simulation runs get very high weight because of differences in outcome- and effect scale across simulation runs (due to the used exponential specification). We therefore generally find average results from the IHDP benchmark difficult to interpret, especially because it is unclear whether improvements occur in all runs or only in these few ‘special’ runs.
> - Second, the IHDP dataset is very small (n=747, with only 139 treated units), which may mean that *any* additional regularization is useful.
> - Third, the treatment assignment bias is indeed different in our simulations than in the IHDP benchmark: distributional differences across treatment groups were artificially introduced into the benchmark dataset originating from the Infant Health and Development Program – a randomized experiment targeting an intervention which provides specialist child care to premature infants with low birth weight – by removing a non-random subset of observations from the treatment group (those with non-white mothers).  The outcomes were then simulated according to exponential-linear response surfaces, where covariates are randomly selected into the linear predictor with 40% probability in each run. It is thus unlikely that there is much alignment between how the outcomes are simulated and how the distributional differences arise; this selection setting may thus be closest to our ‘non-confounded propensity’ specification.
> - Finally, let us briefly outline in what kind of scenario we would expect CFRNet’s balancing regularization to be most likely to work: the underlying principle is analogous to domain-adversarial training from the domain adaptation literature, which is often motivated by the idea of removing *spurious* correlations/cues. We would therefore expect CFRNet to work best when treatment is selected based on features that (i) do not affect outcome themselves but (ii) are correlated with features that do affect outcome (leading to possible spurious reliance on these features in unregularized models). It may be that the IHDP DGP is closer to such a setting.
>
> ### References in the response
>
> [R1] Schwab, Patrick, et al. "Learning counterfactual representations for estimating individual dose-response curves." Proceedings of the AAAI Conference on Artificial Intelligence. Vol. 34. No. 04. 2020.
>
> [R2] Kaddour, Jean, et al. "Causal effect inference for structured treatments." Advances in Neural Information Processing Systems 34 (2021): 24841-24854.
>
> [R3] Yoon, Jinsung, et al. "GANITE: Estimation of individualized treatment effects using generative adversarial nets." International Conference on Learning Representations. 2018.
>
> [R4] Zhang, Yao, et al. "Learning overlapping representations for the estimation of individualized treatment effects." International Conference on Artificial Intelligence and Statistics. PMLR, 2020.
>
> [R5] Johansson, Fredrik, Uri Shalit, and David Sontag. "Learning representations for counterfactual inference." International conference on machine learning. PMLR, 2016.
>
> [R6] Curth, Alicia, and Mihaela van der Schaar. "On inductive biases for heterogeneous treatment effect estimation." Advances in Neural Information Processing Systems 34 (2021): 15883-15894.
>
> [R7] Shalit, Uri, et al. "Estimating individual treatment effect: generalization bounds and algorithms." International Conference on Machine Learning. PMLR, 2017.
>
> [R8] Shi, C., Blei, D., & Veitch, V. (2019). Adapting neural networks for the estimation of treatment effects. Advances in neural information processing systems, 32.
>
> [R9] Curth, A., Svensson, D., Weatherall, J., & van der Schaar, M. (2021, August). Really doing great at estimating CATE? a critical look at ML benchmarking practices in treatment effect estimation. In Thirty-fifth Conference on Neural Information Processing Systems Datasets and Benchmarks Track (Round 2).

---

> > ### Comment · Reviewer_MjU8 · 2022-08-18
> > **Thank you for the response**
> >
> > Thank you for the detailed explanation. It clarifies the issue around CFRNet.

---

> ### Author Response · Authors · 2022-08-14
> **Response to Reviewer MjU8: Part 1/2**
>
> Thank you for your thoughtful comments and suggestions. We are grateful for your appreciation of our paper! We respond to your major points in turn below.
>
> ### Simulations based on large-scale datasets.
>
> Allow us to emphasize that in our experiments we use datasets that are standard throughout the ML literature for CATE estimation, ranging in size from 2200 (ACIC2016) to 11400 examples (Twins). The features from the datasets we used in our benchmark have been commonly used in semi-synthetic set-ups for evaluation by several CATE estimation methods: TCGA [R1, R2], Twins [R3, R4], News [R1, R5], ACIC2016 [R6]. In fact, the Twins dataset is one of the larger-scale datasets (in terms of the number of data points) commonly used throughout the literature.
>
> In this paper, we are proposing a new dimension for evaluating CATE models in terms of their ability to discover the correct predictive covariates when estimating the causal effects. The benchmark environment that we have developed for evaluating this new dimension can be easily extended to incorporate additional datasets and models. In particular, our proposed Data Generating Process (DGP) can be used with any covariates dataset as input (see Algorithm 1). We have now also updated the documentation for our code to provide instructions for adding a new dataset and/or CATE model such that users can easily evaluate their CATE methods in new scenarios.
>
> Moreover, as per your suggestion, we have now performed additional experiments where we have increased the scale of our datasets in terms of the dimensionality of the feature space. In particular, we have obtained a version of the TCGA data which has the $5000$ most variable genes as features (instead of 100 as used in our initial experiments). Out of the $d=5000$ features, we set $n_{\mathcal{I}} = 500$ of them to be important as prognostic and predictive for each potential outcome. We report here:  https://imgur.com/a/92AErOd the performance comparison in terms of $Attr_{pred}$ (left, higher is better), $Attr_{prog}$ (middle, lower is better) and PEHE (right, lower is better) when varying the predictive scale. We have also included a new section in the paper to discuss the results in Appendix C.4. We find that in this setting,  the different learners have poor absolute performance in discovering the predictive features, while relative performance remains similar to the smaller scale experiments. Thus, our benchmark suggests that developing CATE learners that can better identify predictive covariates in a high dimensional setting is needed and would represent an interesting direction for future work.
>
> ### Hyperparameter tuning in the experiments.
>
> Hyperparameter selection for methods for CATE estimation is particularly challenging because counterfactuals are never observed in real data (i.e. we can only observe one of the potential outcomes for each patient). Because of this, it would be unfair to use the counterfactuals simulated with our DGP for hyperparameter tuning as it would not be possible to use such an approach for real datasets. Alternative ways of selecting hyperparameters that have been used throughout the literature involve using only the error on the factual outcomes [R2, R3] or using a nearest-neighbour approach for constructing a counterfactual on the test set [R1, R4, R7].
>
> Please note that in our experiments, we have used 30 different random seeds for each of the 4 datasets and for each setting of the predictive scale $\omega_{\mathrm{pred}} \in \{10^{-3}, 10^{-2}, 10^{-1}, 0.5, 1\}$ and non-linearity scale $\omega_{\mathrm{nl}} \in \{0.0, 0.2, 0.5, 0.7, 0.9, 1.0\}$  and 10 different random seeds for each of the 4 datasets, for each of the predictive confounding, prognostic confounding  and non-confounded propensity settings when changing the $\omega_\pi \in \{0, 0.5, 1, 2, 5, 10\} $. Given the scale of our experiments, which consists of 4 x 30 x 5 + 4 x 30 x 6 + 10 x 4 x 3 x 6 = 2040 experimental runs in total,  performing hyperparameter tuning for each one of them would have been computationally prohibitive.
>
> The way we have chosen hyperparameters for the different CATE models was to ensure that the different models had similar capacities, thus ensuring a fair comparison across our different experimental settings. Such an approach has also been used by [R6, R8] to investigate the performance of CATE models under many different experimental settings. Note that for methods such as CFRNet, where the  discrepancy-based
> regularization term controlled by hyperparameter $\gamma$ plays a significant role in performance, we have reported results for various different settings of this hyperparameter.

---

> > ### Comment · Reviewer_MjU8 · 2022-08-18
> > **Thank you for answer**
> >
> > Thank you for your helpful and extensive answer on both points.
> > ### Large scale data
> > I appreciate you new high-dimensional setting and suppose it to be valuable for the paper. Besides increasing the dimensionality, I thought about using datasets of larger sample sizes, so that the confidence intervals will be smaller and you will be able to discover significance in those experiments where this was not the case on smaller data. For the large-scale datasets, it might be worth to look for the open-source datasets popular in Uplift Modeling (similar to heterogeneous treatment effect estimation, yet more business oriented problem, appearing in online advertising, credit scoring, etc), like Hillstrom Email Marketing data [1] (60k points) or Criteo Uplift Modeling dataset [2] (14M points).
> > ### Hyperparameter tuning
> > Your comment sounds reasonable and convincing for me. My only remark is that there are still some metrics or technics to perform model selection for CATE, such as policy risk [3] (evaluating treatment assignment rule), counterfactual cross-validation [4] or influence function based selection [5]. What do you think about applying them in your benchmark?
> >
> > #### References
> > [1] Kevin Hillstrom. The MineThatData e-mail analytics and data mining challenge. http://www.minethatdata.com/Kevin_Hillstrom_MineThatData_E-MailAnalytics_DataMiningChallenge_2008.03.20.csv, March 2008.
> >
> > [2] Eustache Diemert, Artem Betlei, Christophe Renaudin, and Massih-Reza Amini. A large scale benchmark for uplift modeling. In Proceedings of the AdKDD’18, 2018.
> >
> > [3] Uri Shalit, Fredrik D Johansson, and David Sontag. Estimating individual treatment effect: generalization bounds and algorithms. In International Conference on Machine Learning, pages 3076–3085. PMLR, 2017.
> >
> > [4] Yuta Saito and Shota Yasui. Counterfactual cross-validation: Stable model selection procedure for causal inference models. In International Conference on Machine Learning, pages 8398– 8407. PMLR, 2020.
> >
> > [5] Ahmed Alaa and Mihaela Van Der Schaar. Validating causal inference models via influence functions. In International Conference on Machine Learning, pages 191–201. PMLR, 2019.

---

> > > ### Author Response · Authors · 2022-08-29
> > > **Thank you for your response!**
> > >
> > > Thank you for your response, we are delighted to hear that you found our answers helpful! Below, we respond to the two additional comments made above:
> > >
> > > - _Large scale data:_ Thank you for pointing us towards these uplift modeling datasets, they are new to us! It would definitely be interesting to extend our benchmarking results in future work by incorporating such datasets with more records to investigate the effect of the amount of training data on covariate discovery.
> > > - _Hyperparameter tuning:_ We think that it would definitely be an interesting avenue for future work to use our benchmark and our new evaluation metric to systematically evaluate the (dis)advantages of the different model selection approaches listed in the review! We also believe that our code can easily be extended to incorporate such model selection metrics to decide the hyperparameters for the CATE learners. However, as we have previously mentioned, performing hyperparameter tuning would make the experiments significantly more computationally prohibitive. We will incorporate a discussion about this in the camera-ready version of the paper!

---

### Meta-Review · Area_Chair_b1wd · 2022-09-13

**Recommendation:** Accept
**Confidence:** 2

**Metareview:**

This submission proposes a semi-synthetic dataset to estimate conditional average treatement estimation (CATE) and an empirical study of recovery of important parameters by CATE estimators.

The submission generated much discussion between the reviewers and the authors. The question and the empirical evidence was seen as interesting. Although it does not isolate factors of a causal-inference scenario that modulate the success of identifying important features for CATE models, it was seen as setting a promising line of research.

---

### Decision · Program_Chairs · 2022-09-16

Accept